# The RNA-binding protein ARPP21 controls dendritic branching by functionally opposing the miRNA it hosts

Frederick Rehfeld [1], Daniel Maticzka[2], Sabine Grosser[3], Pina Knauff[1], Murat Eravci[4], Imre Vida [3], Rolf Backofen [2] & F. Gregory Wulczyn [1]

About half of mammalian miRNA genes lie within introns of protein-coding genes, yet little is known about functional interactions between miRNAs and their host genes. The intronic miRNA miR-128 regulates neuronal excitability and dendritic morphology of principal neurons during mouse cerebral cortex development. Its conserved host genes, *R3hdm1* and *Arpp21,* are predicted RNA-binding proteins. Here we use iCLIP to characterize ARPP21 recognition of uridine-rich sequences with high specificity for 3'UTRs. ARPP21 antagonizes miR-128 activity by co-regulating a subset of miR-128 target mRNAs enriched for neurodevelopmental functions. Protein–protein interaction data and functional assays suggest that ARPP21 acts as a positive post-transcriptional regulator by interacting with the translation initiation complex eIF4F. This molecular antagonism is reflected in inverse activities during dendritogenesis: miR-128 overexpression or knockdown of ARPP21 reduces dendritic complexity; ectopic ARPP21 leads to an increase. Thus, we describe a unique example of convergent function by two products of a single gene.

---

[1] Institute for Cell Biology and Neurobiology, Charité-Universitätsmedizin Berlin, Corporate Member of Freie Universität Berlin, Humboldt-Universität zu Berlin, and Berlin Institute of Health, Charitéplatz 1, 10117 Berlin, Germany. [2] Department of Computer Science, Albert-Ludwigs-Universität Freiburg, Georges-Köhler-Allee 106, 79110 Freiburg im Breisgau, Germany. [3] Institute for Integrative Neuroanatomy, Charité-Universitätsmedizin Berlin, Corporate Member of Freie Universität Berlin, Humboldt-Universität zu Berlin, and Berlin Institute of Health, Charitéplatz 1, 10117 Berlin, Germany. [4] Institute for Chemistry and Biochemistry, Freie Universität Berlin, Thielallee 63, 14195 Berlin, Germany. These authors contributed equally: Daniel Maticzka, Sabine Grosser, Pina Knauff. These authors jointly supervised this work: Imre Vida, Rolf Backofen. Correspondence and requests for materials should be addressed to F.G.W. (email: gregory.wulczyn@charite.de)

Multiple mechanisms act at the post-transcriptional level to enable the dynamic adjustment of gene expression; an ability critical to the correct specification and differentiation of cell types during development. Among these, miRNAs represent an important class of non-coding RNAs that bind to partially complementary sites in target mRNAs to inhibit their translation and accelerate their decay[1]. The initial primary transcripts of miRNAs undergo two sequential endonucleolytic processing steps by the RNase type III enzymes Drosha and Dicer. The resulting ≈22-nucleotide (nt) mature miRNA is incorporated into the miRNA-induced silencing complex responsible for the selection, binding, and repression of target transcripts[2]. Because a single miRNA species can inhibit hundreds of different target transcripts, miRNAs are able to shape gene expression at a global level[3]. Brain-specific knockouts of core miRNA pathway genes established the importance of miRNAs for neural development[4]. Subsequent studies of individual miRNAs have revealed functions in almost all aspects of neurodevelopment: from the specification of neural fate[5] to migration[6,7], dendritic growth[8], and synapse formation[9].

Several studies have investigated the roles of miR-128 in mouse brain development, including the control of progenitor proliferation and differentiation[10–12]. Loss of miR-128 function in mice causes neuronal hyperexcitability accompanied by severe and lethal seizures[13]. During cortex development miR-128 inhibits migration and limits dendritic growth and complexity of upper-layer neurons[6]. The effects of miR-128 overexpression on migration and dendritic complexity could be rescued by co-expression of one of its regulatory targets, the intellectual disability syndrome gene *Phf6*[6,14]. A better understanding of how miR-128 activity is regulated would provide insight into how miR-128 can perform its multiple developmental functions.

Mammals have two genes for miR-128 that are located in introns of two conserved, orthologous protein-coding host genes, *R3hdm1* and *Arpp21*, that harbor miR-128-1 and miR-128-2, respectively. This arrangement is evolutionary conserved (Supplementary Table 1a) and implies transcriptional coupling of the host genes and their respective miRNA precursors. The two miR-128 isoforms differ in respect to their stem-loop precursor sequences but yield identical mature 22-nt RNAs[15]. Approximately 40% of all miRNAs reside in introns[16], but few examples of functional connections between intronic miRNAs and their host genes exist[17–19]. ARPP21 is upregulated in mouse miR-128 loss-of-function mutants[13], most likely through a conserved binding site for miR-128 in the *Arpp21* 3′ untranslated region (UTR; Supplementary Table 1b; Supplementary Fig. 1a). Most importantly, R3HDM1 and ARPP21 are each members of a family of uncharacterized putative RNA-binding proteins (RBPs) related to the *Drosophila* Encore protein.

Focusing on ARPP21, we show that the conserved R3H and SUZ domains located in the N terminus of full-length isoforms of the protein mediate RNA-binding. The extended C terminus contains an independent transactivation domain, consistent with the ability of this domain to physically interact with the eukaryotic translational initiation factors 4A and 4G (eIF4A and eIF4G). Using individual-nucleotide resolution crosslinking and immunoprecipitation (iCLIP) to characterize mRNA substrates for ARPP21, we found that ARPP21 preferentially binds to uridine-rich sequences in the 3′ UTRs of mRNAs. ARPP21 binds and transactivates the mRNAs for a number of well-characterized targets for miR-128-mediated silencing, including *Phf6*. Consistent with this, ARPP21 overexpression and knockdown experiments show that ARPP21 is a positive regulator of dendritic growth. Together, our results describe an unprecedented antagonistic molecular and functional relationship between a novel RBP and the miRNA it hosts.

## Results

**Developmental regulation of the miR-128 host genes.** Nearest-neighbor analysis of the *Drosophila* translational regulator *encore* (DmeI∖enc) gene family suggests *R3hdm1* and *Arpp21* were generated by duplication from an *encore*-like ancestral gene (Supplementary Fig. 1b). The gene structure of the two mouse miR-128 host genes is shown in Fig. 1a. Like Encore, R3HDM1 and ARPP21 possess adjacent N-proximal R3H and SUZ domains and a large, unstructured C terminus (Fig. 1b). The positively charged R3H domain contains a conserved R-X-X-X-H sequence motif (Fig. 1c) that is predicted to bind RNA[20]. Likewise, the SUZ domain of the *Caenorhabditis elegans* protein SZY-20 was shown to mediate RNA-binding[21]. In addition to the full-length protein, *Arpp21* encodes a number of splice variants, including a truncated version consisting of the initial 88 amino acids (aa) that lacks the R3H and SUZ domains (Fig. 1a, b). Since the *Arpp21* gene accounts for approximately 80% of mature miR-128[13] it became the focus of this investigation.

To begin the characterization of ARPP21, we compared its expression pattern in mouse brain development to miR-128. Quantification of miR-128 expression by quantitative reverse transcription-PCR (qRT-PCR) revealed an ≈200-fold increase from embryonic day 12 (E12) to postnatal stages (Supplementary Fig. 1c), confirming an earlier northern blot analysis[22]. *R3hdm1* and the two major *Arpp21* transcript isoforms each showed increased expression over time (Supplementary Fig. 1d). Similarly, in situ analysis of the full-length and truncated *Arpp21* mRNA splice variants at E15.5 revealed accumulation in the postmitotic neurons of the cortical plate and relative absence in the progenitor regions of the ventricular and subventricular zones (Fig. 1d). This closely corresponds to the pattern seen for miR-128 at this stage[6]. The two ARPP21 protein isoforms, however, vary in their expression during brain development. Full-length ARPP21 protein levels steadily increased during mouse brain development (Fig. 1e), comparable to miR-128 (Supplementary Fig. 1c). In comparison, expression of the 88-aa isoform shows a steeper postnatal increase (Fig. 1f). The two ARPP21 isoforms also differ in their tissue distribution, as the short protein variant is absent from the thymus and within the brain is more restricted to cortical, hippocampal, and striatal regions compared to the long form (Fig. 1g). Differential regional expression of the two variants is supported by isoform-specific in situ hybridizations from the Allen Brain Atlas[23] (Supplementary Fig. 1e). Differential regulation of the two isoforms might be related to alternative promoter usage, as indicated by differential patterns of tri-methylated lysine 4 on histone 3 (H3K4me3) peaks that suggest the presence of tissue-specific promoters for each isoform (Supplementary Fig. 1f).

**Localization of ARPP21 and R3HDM1 to stress granules.** We next analyzed the subcellular localization of ectopic ARPP21 and R3HDM1 using FLAG-tagged versions expressed in HeLa cells. Anti-FLAG staining revealed almost exclusively granular, cytoplasmic localization of both full-length proteins (Fig. 2a, b). Occasionally, we also observed cells containing larger cytosolic aggregates that resembled cytosolic stress granules (SGs; Supplementary Fig. 2a). SGs are phase-dense cytosolic aggregates that form upon environmental stress and contain translationally silent mRNAs bound to various proteins, including the 40S ribosomal subunit and several eukaryotic translation initiation factors[24]. To test if R3HDM1 or ARPP21 can be recruited to SGs, HeLa cells were treated with arsenite (Fig. 2c, d). Co-staining for full-length transfected FLAG-tagged R3HDM1 or ARPP21 revealed a high degree of overlap with the SG marker eIF3η (also referred to as EIF3A) in all treated cells (Fig. 2e–g). These results were

confirmed by the use of clotrimazole or heat shock to induce stress (Supplementary Fig. 2b, c) and by co-staining with the additional SG markers FXR2 or G3BP (Supplementary Fig. 2d-f). Using different truncation mutants we found that the short ARPP21 isoform does not localize to SGs and that C-terminal sequences in full-length ARPP21 and R3HDM1 are necessary and sufficient for stress-induced relocalization (Fig. 2h–j; Supplementary Fig. 2g-j). The predicted RNA-binding domains, however, are dispensable, suggesting that protein–protein interactions might be sufficient for recruitment to SGs. We could also demonstrate relocalization of endogenous ARPP21 present in the somatodendritic compartment of primary cortical neurons to eIF3η-positive SGs upon arsenite treatment (Fig. 2k, l; Supplementary Fig. 3).

**ARPP21 and R3HDM1 are post-transcriptional activators.** SGs are subcellular compartments involved in the post-transcriptional regulation of gene expression[24]. To test for regulatory activity of the two host proteins we prepared fusion proteins with the bacteriophage MS2 coat protein to allow tethering to mRNAs containing MS2-specific RNA-binding sites, in this case a green fluorescent protein (GFP) reporter construct harboring four consecutive MS2-binding sites in the 3′ UTR (Fig. 3a)[25]. We also

tethered AGO2, the main miRNA effector protein, as a control for the assay based on its known inhibitory activity in tethering experiments[26]. Whereas AGO2 led to an approximately 3-fold reduction in GFP fluorescence compared to the MS2 protein control, full-length ARPP21-MS2 or R3HDM1-MS2 fusion proteins increased GFP fluorescence between 2-fold and 1.5-fold, respectively (Fig. 3b, c). The stimulatory effect of ARPP21 mapped to the C-terminal sequences from aa 365 to 807 (Arpp21ΔCterm); the R3H and SUZ domains were dispensable for the stimulatory effect (Arpp21ΔR + S) (Fig. 3c). None of the constructs altered GFP reporter expression without fusion to MS2 (Fig. 3d), demonstrating the requirement for RNA tethering.

**Interaction partners for ARPP21 include eIF4A and eIF4G.** We next set out to identify functional interaction partners for ARPP21 and R3HDM1. For this purpose, we immunoprecipitated FLAG-tagged versions of both mouse proteins after transfection of HEK-293T cells (Supplementary Fig. 4a) and identified protein interaction partners by liquid chromatography coupled to tandem mass spectrometry (LC-MS/MS; Fig. 3e). To minimize background and the potential for RNA-bridged interactions, RNase treatment was performed prior to immunoprecipitation (IP). LC-MS/MS analysis identified 150 proteins with ≥2-fold

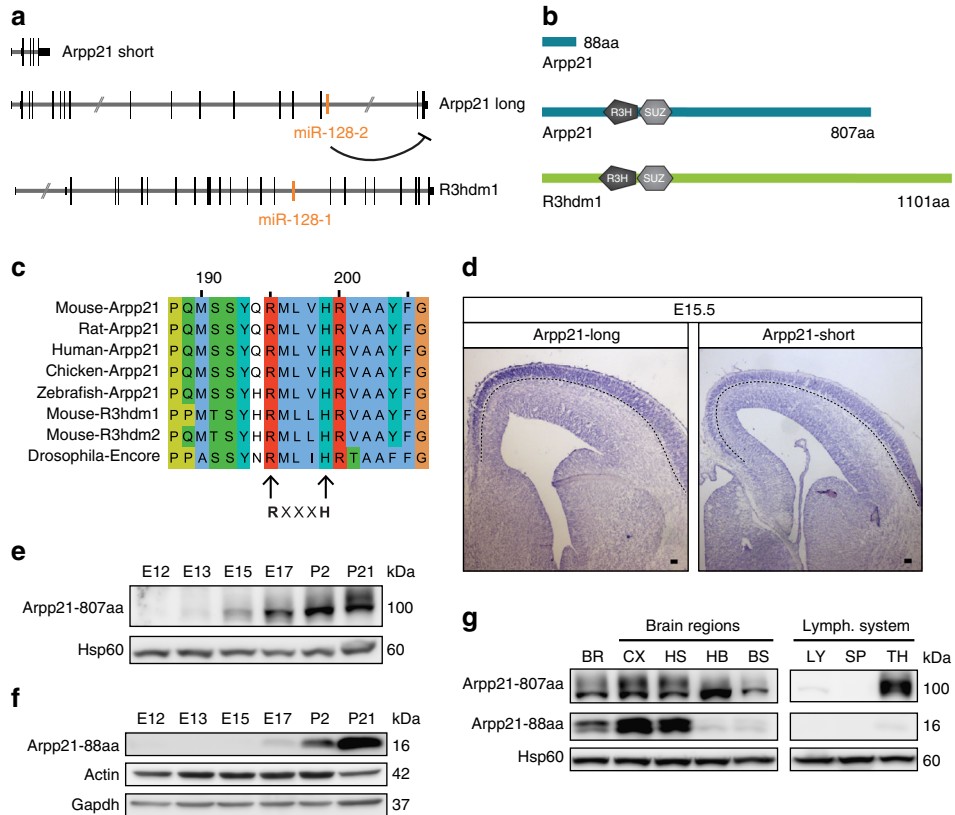

**Fig. 1** miR-128 host gene conservation and ARPP21 expression during brain development. **a** Genomic localization of murine miR-128 within the host genes *Arpp21* and *R3hdm1*. Two major splice isoforms of *Arpp21* are depicted, as referred to in the text (*Arpp21* short and long). **b** Domain topology of ARPP21 and R3HDM1 proteins, including major ARPP21 isoforms (*Arpp21* short = ARPP21-88aa and *Arpp21* long = ARPP21-807aa). **c** Protein sequence alignment of the core R3H motif of ARPP21 from different vertebrate species, murine R3HDM1, murine R3HDM2, and fly ENCORE using the COBALT alignment tool (NCBI). The arginine and histidine residues of the R3H motif are marked by arrows. **d** In situ hybridization of E15 mouse brains with probes specific for *Arpp21*-long and -short transcript isoforms. Expression of both isoforms is enriched in post-mitotic neurons of the cortex. The dotted line marks the lower boundary of the cortical plate. Scale bar: 100 μm. **e**, **f** Detection of ARPP21-807aa and ARPP21-88aa during brain development by western blot. **e** ARPP21-807aa shows a gradual increase in expression. **f** Expression of ARPP21-88aa is strongly increased during postnatal stages. **g** Immunoblot analysis of both ARPP21 protein variants in samples from different brain regions and the lymphatic system of P21 mice. ARPP21-807aa is detected in several brain regions and the thymus. ARPP21-88aa is only detected in the whole brain, cortical, striatal, and hippocampal samples. *BR* whole brain, *CX* cortex, *HS* hippocampus and striatum, *HB* hindbrain, *BS* brainstem, *LY* lymph nodes, *SP* spleen, *TH* thymus. Full western blot images are presented in Supplementary Fig. 21

higher peptide intensities in the ARPP21 and R3HDM1 IPs compared to IPs from mock-transfected cells (Supplementary Table 2). Of these, 31 were more than 2-fold enriched for either ARPP21 or R3HDM1, suggesting that the experiment identified both protein-specific as well as common interaction partners. We focused on shared candidates, reasoning that functionally

important interactions are likely to be conserved between the two homologs. Among these was R3HDM1 itself, as evidenced by the recovery of several human-specific peptides derived from the endogenous HEK-293T protein. Co-immunoprecipitation (co-IP) experiments with differentially tagged versions of R3HDM1 and ARPP21 were performed to verify homomeric interactions. After co-transfection of FLAG-tagged ARPP21 and ARPP21-GFP fusion constructs, GFP-tagged ARPP21 was detected in IPs using anti-FLAG antibody, and FLAG-tagged ARPP21 was recovered after IP with anti-GFP antibody (Fig. 3f). Similar results were obtained for R3HDM1 (Supplementary Fig. 4b).

The most abundant peptides detected by MS were derived from 14-3-3 proteins, a protein family known to bind specific phosphomotifs present in a wide range of cellular substrates. R3HDM1 and ARPP21 are known phosphoproteins[27,28] and co-IP experiments with one of the 14-3-3 proteins detected, 14-3-3-ζ, showed phosphatase-sensitive interaction with the two host proteins (Supplementary Fig. 4c). This result confirms the MS experiment and suggests that 14-3-3 proteins might regulate host protein function.

Most promising with regard to the tethering results were several candidate interactors with known functions in mRNA translation and stability, including eIF4G and eIF4A (Supplementary Fig. 4d). Both are essential components of the eIF4F[29]. eIF4G serves as an interaction platform and binds the polyA-binding protein PABP, the 5′-cap-binding protein eIF4E, and the 43S pre-initiation complex component eIF3. eIF4A represents an additional interaction partner of eIF4G that possesses RNA helicase activity and improves ribosomal 5′-to-3′ scanning by unfolding 5′ UTR secondary structure[29]. IP of FLAG-tagged ARPP21 after transfection of N2A cells retrieved endogenous eIF4G and eIF4A (Fig. 3g). Co-IP was not RNase-sensitive, suggesting the interaction is not RNA-mediated. The interaction with two closely related eIF4A paralogs eIF4A1 or eIF4A2 was tested by co-IP after co-transfection and revealed a substantially stronger co-IP of eIF4A1 with ARPP21 or R3HDM1 compared to eIF4A2 (Fig. 3h, i). To determine the domain in ARPP21 that is responsible for mediating the interaction with eIF4A and eIF4G, the co-IPs were repeated after transfection of cells with ARPP21 deletion mutants consisting of either full-length, N-terminal (ΔC-term), or C-terminal (ΔR3H + SUZ) domains. Interestingly, the same C-terminal unstructured domain present on the ΔR3H +

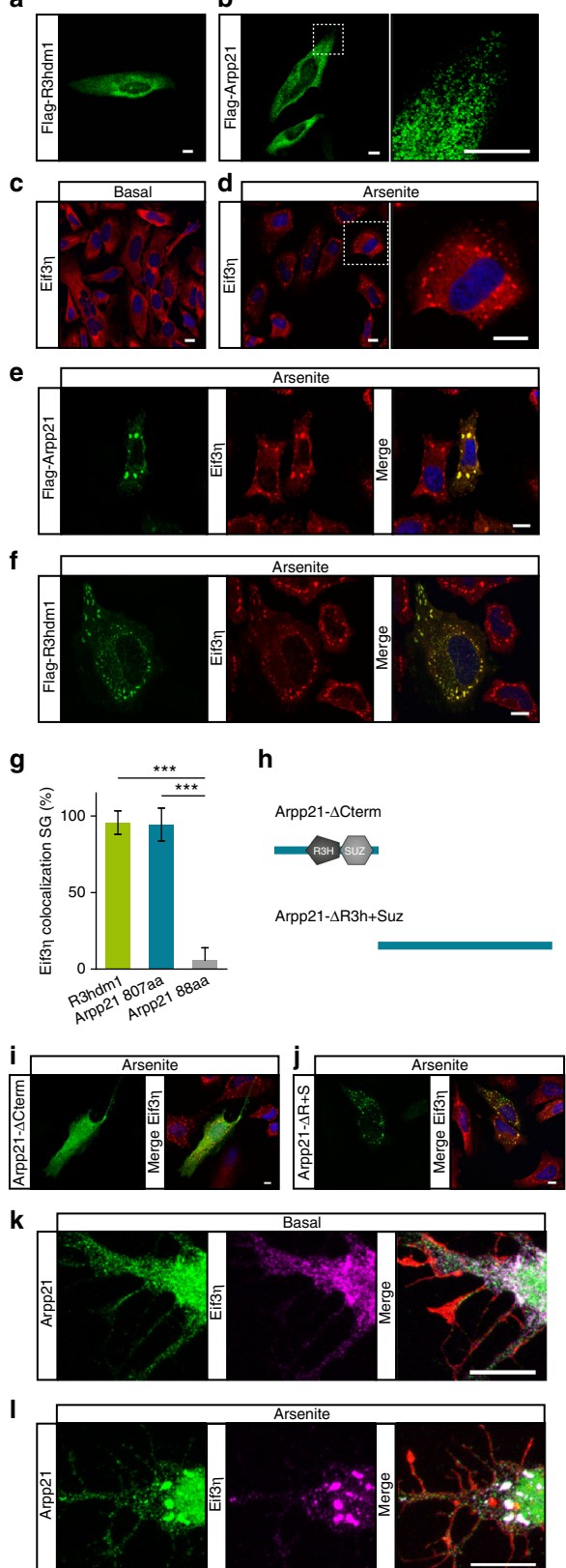

**Fig. 2** R3HDM1 and ARPP21 localize to the cytosol and show overlapping staining with the stress granule marker eIF3η in response to stress. **a**, **b** FLAG immunostainings of HeLa cells expressing N-terminally FLAG-tagged R3HDM1 or ARPP21 reveal cytosolic localization in small, perinuclear-enriched granules. The right panel in **b** shows a higher magnification of the dotted box in the left panel. **c** Diffuse, cytosolic staining of the stress granule marker eIF3η is visible under basal conditions. **d** Treatment of HeLa cells with arsenite induces stress granules positive for eIF3η. **e**, **f** ARPP21 and R3HDM1 colocalize with eIF3η upon arsenite stimulation. **g** Quantification of eIF3η colocalization with R3HDM1 and the two ARPP21 protein isoforms. Data expressed as mean ± s.d. R3HDM1: $n = 94$ cells. ARPP21-807aa: $n = 89$ cells. ARPP21-88aa: $n = 141$ cells. ***$p < 0.001$, Student's $t$-test. **h** Schematic depiction of ARPP21 truncation mutants used in **i** and **j**. **i** ARPP21 lacking its C-terminal region (green) shows no colocalization with eIF3η (red) upon cellular stress. **j** Upon cellular stress localization of the C-terminal ARPP21 construct lacking N-terminal domains (green) overlaps with eIF3η (red). **k**, **l** Staining of endogenous ARPP21 (green), eIF3η (magenta), and phalloidin (red) in cultured primary cortical neurons after 4 days in vitro (DIV4). **k** Under basal conditions ARPP21 localizes to neuronal cell bodies and proximal dendrites. **l** Upon arsenite treatment ARPP21 colocalizes with the SG marker eIF3η. Scale bar in all images: 10 μm

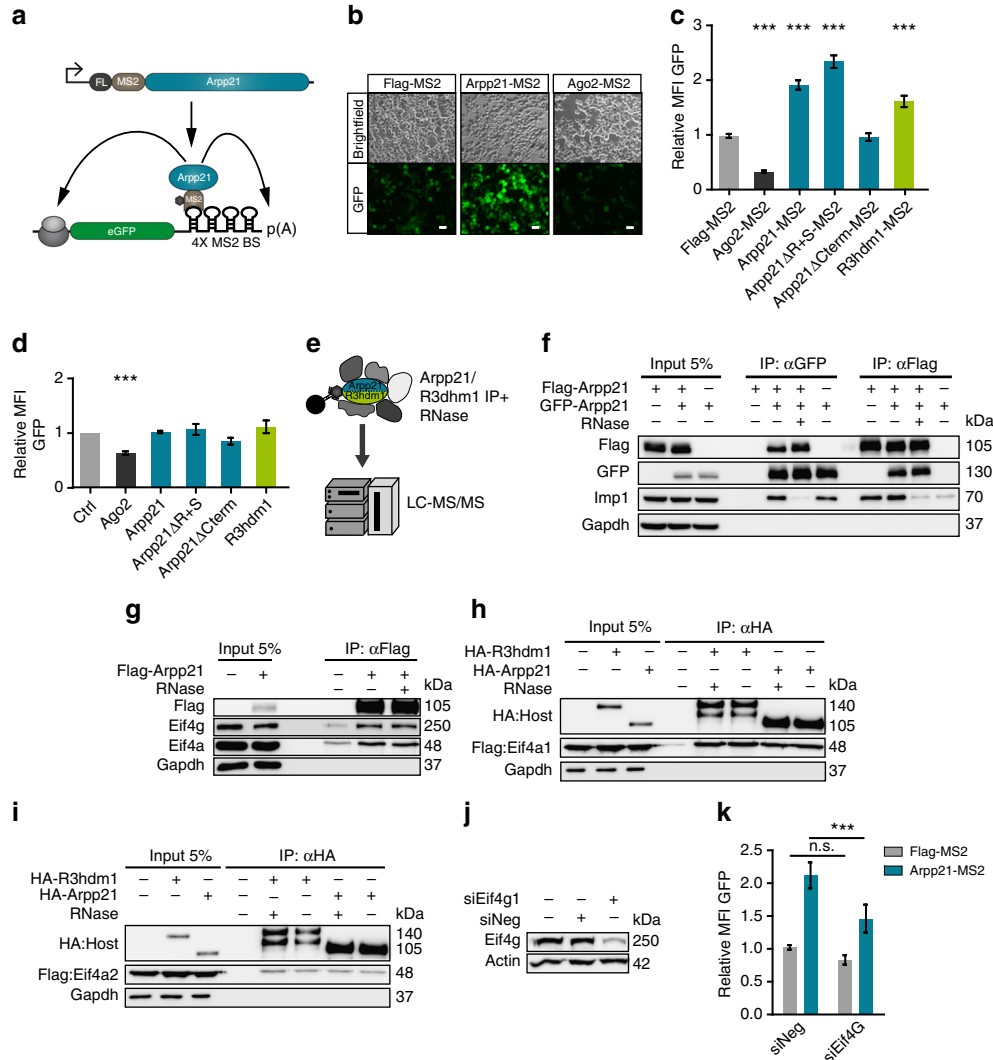

**Fig. 3** ARPP21 acts as a positive regulator of post-transcriptional gene expression in a tethering assay and functionally interacts with parts of the eIF4F complex. **a** The MS2 coat protein was used to tether ARPP21 to the 3′ UTR of a GFP reporter transcript in 293T cells. **b** ARPP21 tethering results in increased GFP fluorescence of the MS2-binding site reporter (4X-MS2-BS-GFP) compared to tethering of the MS2 protein alone. Tethering of the translational inhibitor AGO2 decreases basal GFP fluorescence. Scale bar: 50 μm. **c** Flow cytometry analysis of tethering assays, expressed as mean fluorescent intensity (MFI) of transfected cells. AGO2-MS2 decreases, and ARPP21-MS2 and R3HDM1-MS2 increase the expression of the 4X-MS2-BS-GFP reporter, relative to Flag-MS2 alone. The C-terminal part of ARPP21 is necessary and sufficient for the positive effect on reporter expression. ***$p < 0.001$, Student's $t$-test, $n = 3$, error bars represent ± s.d. **d** Untethered ARPP21 and R3HDM1 have no significant effect on GFP expression from the 4X-MS2-BS-GFP reporter relative to MS2 control. Statistical analysis as in **c**, $n = 3$. **e** FLAG-tagged ARPP21 or R3HDM1 were immunoprecipitated from 293T lysates after RNase digestion and protein interactors were identified by LC-MS/MS. **f** Co-immunoprecipitation (co-IP) using FLAG- or GFP-tagged ARPP21 constructs verify the ability of ARPP21 to self-interact by reciprocal co-IP. RNAse treatment strongly reduces co-IP of the RNA-binding protein IMP1 but not homomeric ARPP21 interactions. Loss of GAPDH protein after IP is shown as control. **g** FLAG-ARPP21 coimmunoprecipitates endogenous eIF4A and eIF4G from N2A cells. Antibodies used for detection of proteins in input and eluate are shown on the left. RNase pretreatment of lysates had no effect on co-IP of eIF4A or eIF4G. **h, i** ARPP21 and R3HDM1 coimmunoprecipitate eIF4A1 more efficiently than eIF4A2. HA-tagged ARPP21 or R3HDM1 was co-transfected with FLAG-tagged eIF4A1 (**h**) or eIF4A2 (**i**); results after immunoprecipitation with anti-HA antibody are shown. **j** Validation of endogenous eIF4G1 knockdown in 293T cells used in tethering assay. **k** ARPP21 activity in the tethering assay is reduced after eIF4G knockdown. ***$p < 0.001$, two-way ANOVA with Bonferroni post-test, $n = 4$, error bars represent s.d. Full western blot images are presented in Supplementary Fig. 21

SUZ construct that was responsible for transactivation activity in the tethering assay was necessary and sufficient for the co-precipitation of eIF4G and eIF4A (Supplementary Fig. 4e). As a control, the same IP was reprobed for 14-3-3-ζ, which displayed the inverse specificity by interacting with the full-length and ΔC-term but not the ΔR3H + SUZ constructs (Supplementary Fig. 4e).

We next tested the effect of eIF4G knockdown on ARPP21 activity in the tethering assay. Acute knockdown of eIF4G (Fig. 3j, Supplementary Fig. 4f) did not significantly alter reporter expression in the MS2 control condition, but significantly reduced the increase in GFP expression upon ARPP21 tethering (Fig. 3k). In all conditions tested, tethering of ARPP21 led to a concomitant increase in steady-state reporter mRNA levels (Supplementary Figure 4g). eIF4G knockdown, on the other hand, did not affect reporter mRNA levels (Supplementary Fig. 4g). This suggests that the translational efficiency and not the stability of the tethered mRNA may be responsible for the observed differences in reporter expression. Together, these results strongly implicate the translational initiation complex

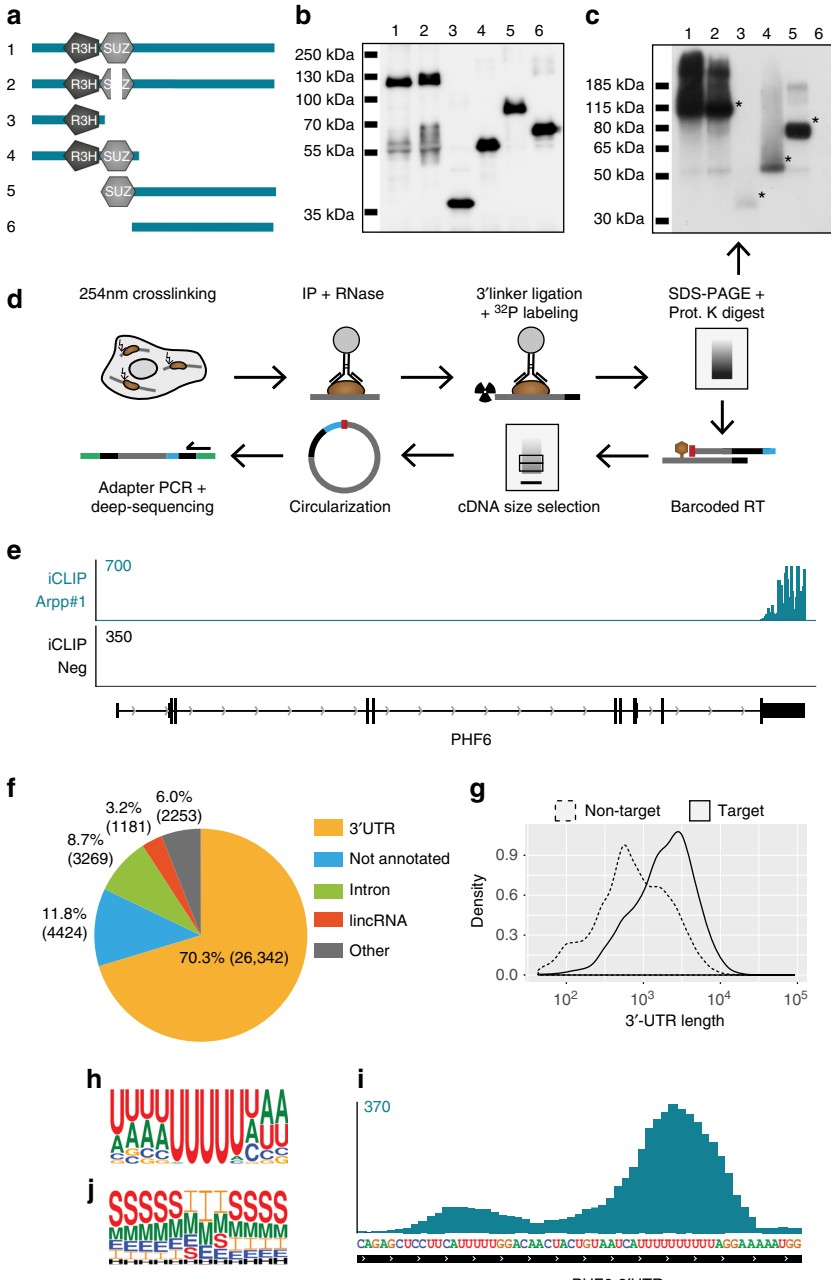

**Fig. 4** iCLIP identifies the ARPP21–RNA interactome. **a** ARPP21 deletion constructs used in UV-crosslinking experiments to assess their RNA-binding potential. Construct 2 conforms to a splice variant that lacks a part of the SUZ domain. **b** SDS-PAGE with FLAG immunoblot confirms immunoprecipitation of all ARPP21 variants, lane designations refer to constructs depicted in **a**. **c** Autoradiogramm of crosslinked and [32]P-labeled ARPP21–RNA complexes separated by NuPAGE, lane designations as in **a** and **b**. All variants harboring at least one of the predicted RNA-binding domains exhibit RNA-binding ability (*), the C-terminal only Construct 6 yielded no signal. **d** Schematic overview of the iCLIP procedure performed with iARPP21 cells. Unique barcodes (blue) introduced during the reverse transcription allow for multiplexing iCLIP samples. Due to the circularization step, truncated cDNAs are captured and the position of protein–RNA crosslinking (red) can be accurately assessed. Green: deep-sequencing adapters. **e** Representative ARPP21 iCLIP coverage at the *Phf6* locus shows specificity for the 3′UTR. The negative control iCLIP from cells not expressing ARPP21 shows negligible background. Numerical scale on left refers to unique crosslinking events (CEs). **f** Genome-wide characterization of unique CEs for ARPP21 by RNA type. **g** 3′UTRs of ARPP21 targets are longer compared to non-ARPP21 target transcripts. Median 3′UTR length of ARPP21 target mRNA: 1.99 kb. Median 3′UTR length of ARPP21-unbound mRNA: 0.72 kb. **h** Nucleotide representation flanking the crosslink site as determined by GraphProt identifies a uridine-rich motif. **i** ARPP21 iCLIP signals are enriched at uridine-rich sequences. A representative peak from the *Phf6* 3′UTR is shown; the numerical scale at left refers to unique CEs. **j** Secondary structure analysis of the ARPP21 interaction motif by GraphProt. The center of the motif containing the crosslinked nucleotide is largely devoid of predicted stems. *E* external loop, *H* hairpin, *I* internal loop, *M* multi loop, *S* stem

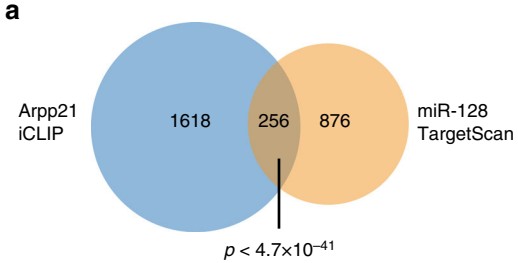

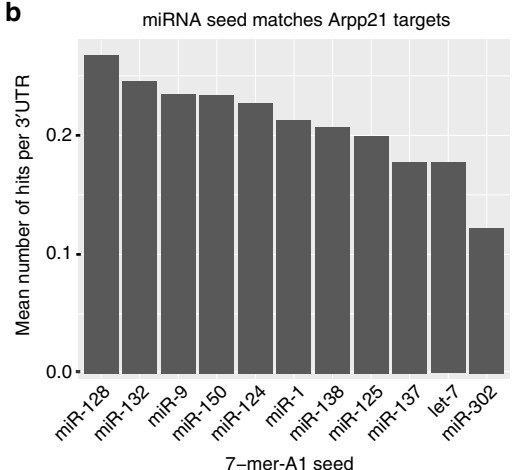

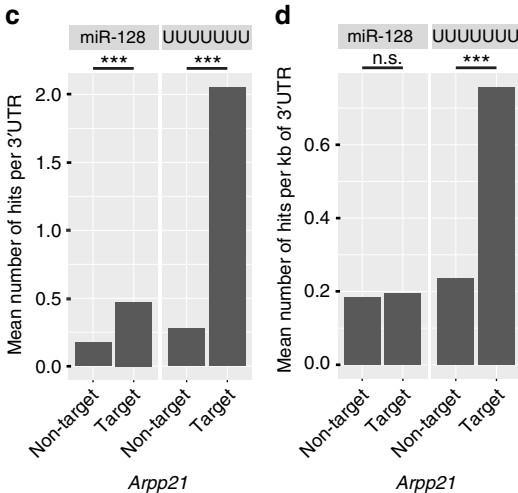

**Fig. 5** ARPP21 and miR-128 bind an overlapping set of target mRNAs with related functions. **a** Venn diagram of conserved TargetScan 7.1 miR-128 target mRNAs, and ARPP21 iCLIP substrate mRNAs. **b** 7mer-A1 seed match occurrence per 3′UTR of ARPP21 targets for several conserved miRNAs. **c** Number of occurrences of miR-128 seed matches and poly-uridine (poly-U) 7mers per 3′UTR among ARPP21-bound and ARPP21-unbound transcripts. Both motifs are significantly enriched in 3′UTRs of ARPP21 target mRNAs. Welch two-sample $t$-test, ***$p < 2.2 \times 10^{-16}$. **d** miR-128 and poly-U 7mer occurrence normalized per kilobase of 3′UTR comparing ARPP21 target and non-target transcripts. ***$p < 2.2 \times 10^{-16}$

eIF4F in mediating post-transcriptional activation by ARPP21, although further work will be required to elucidate the precise mechanism.

**iCLIP reveals ARPP21 binds to uridine-rich motifs in 3′ UTRs.**
The preceding experiments established that ARPP21 has

transactivation activity and identified interaction partners that potentially mediate ARPP21-dependent post-transcriptional regulation. The presence of a C-terminal transactivation domain and two RNA-binding domains near the N terminus suggest that the full-length ARPP21 protein might have a modular function. To study possible RNA-binding activity a doxycycline-inducible TREx-293 cell line (iARPP21 cells) was generated allowing dose-controlled expression of hemagglutinin (HA)-tagged ARPP21 (Supplementary Fig. 5a). To assess RNA-binding, ultraviolet (UV)-crosslinking experiments with iARPP21 cells were performed (Supplementary Fig. 5b). After immunopurification with HA antibody, protein–RNA complexes were visualized by $^{32}$P end-labeling of the RNA followed by SDS-polyacrylamide gel electrophoresis. Crosslinking followed by IP for ARPP21 using high-stringency wash steps to disrupt most protein–protein interactions (see Methods) revealed a distinct autoradiograph signal corresponding to the expected size of ARPP21–RNA complexes. The signal was sensitive to RNase I treatment prior to labeling and was absent in IPs from a control TREx-293 cell line that does not express detectable levels of ARPP21 (Supplementary Fig. 5b). To map the regions of ARPP21 necessary for RNA-binding a series of deletion mutants were transiently transfected and analyzed after crosslinking (Fig. 4a). Immunoblotting confirmed similar IP efficiency for each construct (Fig. 4b). Although the signal intensity varied considerably, RNA-binding was detected for all variants harboring at least one of the two predicted RNA-binding domains, R3H or SUZ (Fig. 4c). In contrast, no signal was obtained with a construct lacking the R3H and SUZ domains, suggesting the C-terminal transactivation domain has little to no independent RNA-binding activity under these conditions. We next used the iARPP21 cells to perform iCLIP experiments to map ARPP21-binding sites (Fig. 4d). A total of four independent ARPP21 iCLIP experiments from two independent iARPP21 cell lines were performed and compared to two independent experiments with control TREx-293 cells (Supplementary Fig. 5c, see Methods for details). Details regarding mappable reads (88 million), unique crosslinking events (24 million), and coverage (average PCR duplicates <4) compared to control are provided in Supplementary Table 3. The reproducibility between the four replicates was high, as indicated by the peak profiles of representative transcripts (Supplementary Fig. 5d). Peak calling identified 2009 ARPP21 target transcripts with iCLIP signals preferentially found within the 3′ UTR of transcripts as exemplified for *Phf6* (Fig. 4e) and quantified in Fig. 4f and Supplementary Table 4 (3′ UTRs >70%; introns <9%; lincRNAs <4%). ARPP21-binding enrichment for 3′UTRs is further indicated by the much larger median 3′UTR length of ARPP21 target mRNAs compared to unbound mRNAs (1.99 versus 0.72 kb, respectively, Fig. 4g). To analyze sequence features in the vicinity of ARPP21-crosslinked sites, we used GraphProt[30], which is a machine-learning approach specifically tailored to predict binding preferences from CLIP data. This analysis revealed a uridine-rich sequence motif centered at the crosslinked nucleotide (Fig. 4h), where the central 4–6 nucleotides show the highest preference for uridine and flanking sequences are over-represented for uridine or adenine. A representative peak from the *Phf6* mRNA is shown at nucleotide resolution in Fig. 4i. However, uridine-rich motifs located within the coding regions of bound mRNAs are generally not recognized, and even within 3′ UTRs are not always occupied by ARPP21 (Supplementary Fig. 5e). To assess the contribution of local structural features to ARPP21-binding, the GraphProt analysis pipeline was used to compute the likelihood of secondary structure close to the crosslinked nucleotide. This revealed a strong preference for single-stranded conformations at the center of the motif (Fig. 4j).

With increasing distance from the center of the motif, the preference for single-stranded conformations decreases.

**Common targets but opposing functions of ARPP21 and miR-128.** We next performed Gene Ontology (GO) and Kyoto Encyclopedia of Genes and Genome (KEGG) pathway analyses to deduce functional information from the ARPP21 mRNA targets we obtained. Even though the iARPP21 cells used for the iCLIP represent a heterologous system, several significantly enriched GO categories like cell adhesion, cell division, and mRNA processing or splicing could be identified (Supplementary Fig. 6a). Significantly enriched KEGG pathways included mRNA surveillance, a pathway involving nonsense-mediated decay (NMD) that was previously shown to be regulated by miR-128[11] (Supplementary Fig. 6b, c; Supplementary Data 1). Other enriched KEGG terms with known links to miR-128 function were transforming growth factor-β (TGF-β)[31] and neurotrophin signaling pathways that may be related to the dendritogenesis and hyperexcitability phenotypes of miR-128[6,13]. These results prompted us to compare the KEGG pathways for ARPP21 targets with that of 1132 mRNAs predicted to harbor conserved miR-128 binding sites as determined by the TargetScan 7.1 database[32]. This revealed a number of similarities in the enrichment pattern, including neurotrophin and TGF-β signaling[13] (Supplementary Fig. 6d; Supplementary Data 2). The overlap between transcripts harboring miR-128 and ARPP21-binding sites (256) was significantly higher than predicted by chance (Fig. 5a) and enrichment analysis performed on this set of transcripts recovered similar pathways, including mRNA surveillance, neurotrophin, TGF-β, and MAP-kinase signaling (Table 1; Supplementary Fig. 6e; Supplementary Data 3). These results suggest that at least some of the regulatory activities of ARPP21 and miR-128 may converge on a shared set of target genes.

We also investigated the possible interface between ARPP21 targets and neuronal miRNA networks by calculating the relative abundance of 7mer (A1) seed matches for a panel of conserved miRNAs in the 3′UTRs of ARPP21 target mRNAs (Supplementary Table 5). The miRNAs chosen include highly conserved and ubiquitous miRNAs (let-7, miR-125) as well as representative tissue-specific miRNAs from stem cells (miR-302), muscle (miR-1), blood (miR-150), and nervous system (miR-9, miR-124, miR-128, miR-132, and miR-138). Of these, the miR-128 seed match was the most abundant, followed by the other brain-enriched miRNAs (Fig. 5b). We then compared the number of miR-128 seeds between ARPP21-bound and -unbound mRNAs, (Supplementary Data 4, 5). The 3′UTRs of ARPP21 target mRNAs harbor a significantly higher number of miR-128 7mer seed

sequences compared to non-target mRNAs (Fig. 5c, left panel). Consistent with the GraphProt results, poly-U 7mers were highly enriched among the ARPP21 target mRNAs (Fig. 5c, right panel). After normalizing for UTR length, enrichment for the poly-U motif but not miR-128 sites retains statistical significance (Fig. 5d). This suggests that the motif is a valid proxy for ARPP21-binding, but the association with miR-128 sites may be subject to more complex co-dependency between 3′UTR length and miRNA targeting (reviewed in ref. [33]).

The substantial overlap between transcripts with binding sites for miR-128 and ARPP21 was surprising, given their opposite activities in post-transcriptional gene expression. To validate the iCLIP results, we therefore focused on predicted ARPP21 targets with known functions downstream of miR-128 in the nervous system. We selected *Phf6*, *Msk1*, *Creb1*, *Upf1*, and *Casc3* for further analysis; their iCLIP signals and conserved miR-128-binding sites are described in Fig. 6a, b and Supplementary Fig. 8a. PHF6 was previously shown to be targeted by miR-128 during cortical neuron migration and dendritogenesis[6]. MSK1 and CREB1 are part of the activity-dependent signaling cascade that is perturbed in miR-128-knockout animals[13,34]. CASC3, MSI2, and UPF1 are components of the NMD pathway subject to inhibition by miR-128 during neuronal differentiation[11]. To independently verify the iCLIP results for these mRNAs RNA-immunoprecipitation (RIP) was performed using HEK-293T cells transiently expressing FLAG-tagged ARPP21 (Supplementary Fig. 7a). Co-purification of each mRNA was quantified by qRT-PCR. All of the selected mRNAs were significantly enriched upon ARPP21 RIP compared to control IPs and to three control RNAs without detectable iCLIP signal used as standards (*Gapdh*, *Rpl27*, 18S rRNA; Supplementary Fig. 7b, c). The relative enrichment ranged from 3-fold for *Upf1* up to 35-fold for *Phf6*.

The 3′UTRs of these transcripts were cloned in GFP reporter constructs to assess direct regulatory effects (Fig. 6c). The relative fluorescence of each reporter was measured by flow cytometry in response to exogenous miR-128 and ARPP21, either alone or in combination. miR-128-mediated silencing was observed for all the 3′UTR constructs, confirming previously published results[6,11,34] or TargetScan predictions[32]. Results for *Phf6* and *Msk1*, the strongest miR-128 targets, are in Fig. 6d, e, all others in Supplementary Fig. 8b. Exogenous ARPP21 had the opposite effect as miR-128 and significantly increased GFP expression, consistent with the results obtained with the tethering assay. As a negative control we chose *Msk2*, a predicted target of miR-128 that is expressed at comparable levels to its paralog *Msk1* in iARPP21 cells but without detectable binding in the iCLIP experiment (Supplementary Data 4). The *Msk2* 3′UTR reporter was significantly downregulated upon miR-128 overexpression, but was not affected by ARPP21 expression (Supplementary Fig. 8c). To test the functional relevance of the predicted ARPP21 interaction motif we identified and deleted an approximately 90-nt-long uridine-rich stretch from a murine *Upf1* 3′UTR reporter construct (Supplementary Fig. 9a). This significantly impaired transactivation of the deletion mutant compared to the wild-type construct by ARPP21 but did not affect either basal expression or suppression by miR-128 (Supplementary Fig. 9b, c). We also confirmed that deletion of either the R3H and SUZ RNA-binding domains or the C-terminal transactivation domain substantially reduced transactivation of 3′UTR reporter constructs derived from target mRNAs (Supplementary Fig. 10a-d).

The reporter results were confirmed for endogenous transcripts in HEK-293T cells. Transfection of a synthetic miR-128 mimic led to reduced protein levels of PHF6 and MSK1, ectopic ARPP21 had the opposite effect. Co-expression of ARPP21 and miR-128 in this assay led to intermediate protein expression (Fig. 6f–h). Comparably, doxycycline treatment of iARPP21 cells resulted in

**Table 1 Enriched KEGG pathway terms in intersection of miR-128 and Arpp21 target genes**

| # | Pathway name | Adj. *p*-value |
|---|---|---|
| 1 | **TGF-β signaling pathway** | 0.0007 |
| 2 | Chronic myeloid leukemia | 0.0102 |
| 3 | Colorectal cancer | 0.0187 |
| 4 | **Neurotrophin signaling pathway** | 0.0187 |
| 5 | Sphingolipid metabolism | 0.0218 |
| 6 | MAPK signaling pathway | 0.0249 |
| 7 | Endometrial cancer | 0.0310 |
| 8 | mRNA surveillance pathway | 0.0310 |
| 9 | Glycerophospholipid metabolism | 0.0310 |
| 10 | Prostate cancer | 0.0328 |

Categories known to be regulated by miR-128 based on published data are marked in red.
Pathways that are present in all three populations are marked bold

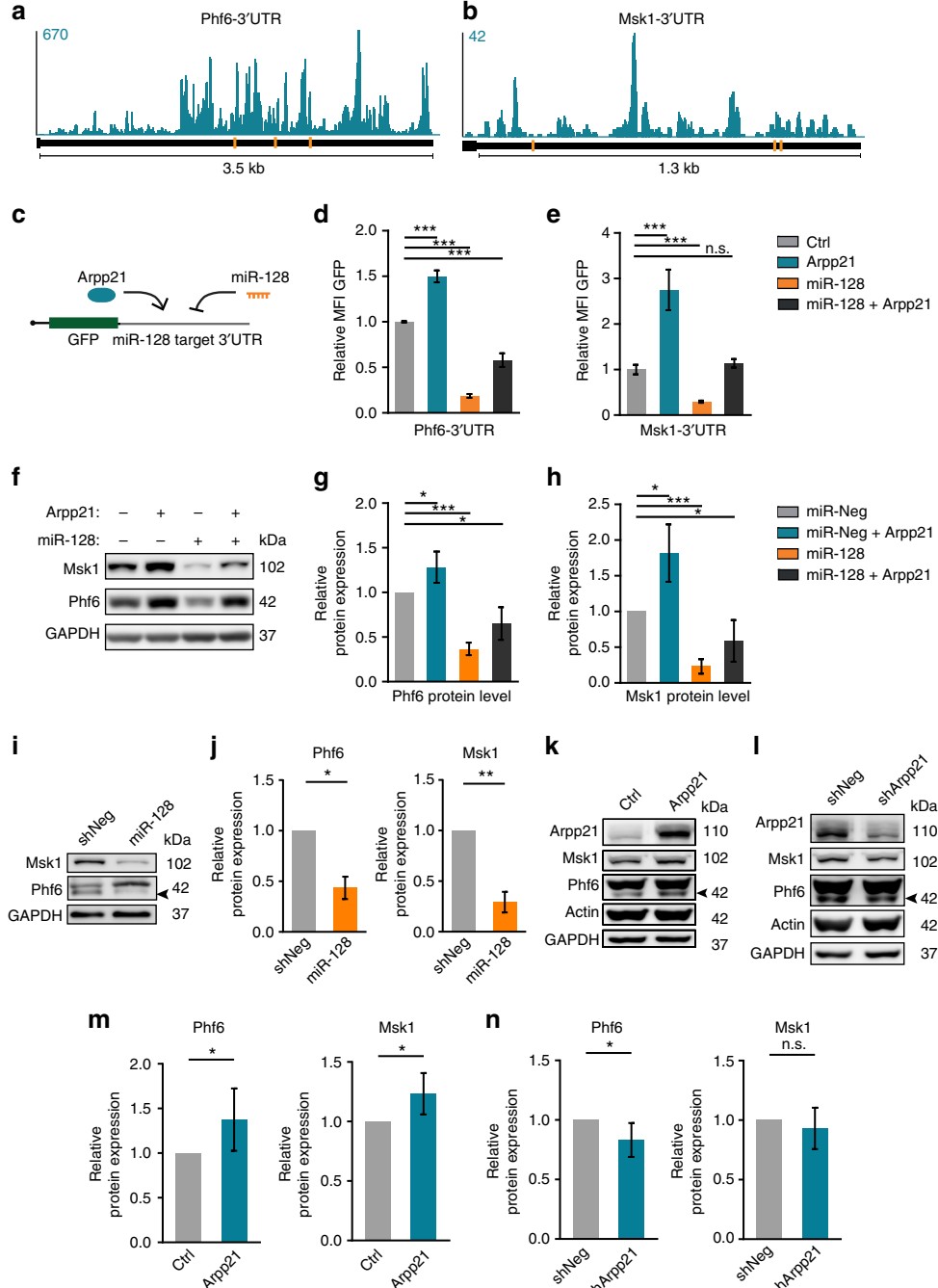

**Fig. 6** ARPP21 and miR-128 have antagonistic functions on an overlapping set of target mRNAs. **a**, **b** Conserved TargetScan predictions for miR-128-binding sites (orange bars) and ARPP21 iCLIP CL signals (turquoise) at the 3′UTRs of the miR-128 target mRNAs *Phf6* and *Msk1*. **c** Schematic of 3′UTR GFP-reporter assay to assess direct effects of miR-128 and ARPP21 on individual transcripts. **d**, **e** *Phf6* and *Msk1* 3′UTR reporter fluorescence upon miR-128 and/or ARPP21 expression normalized to control transfection. miR-128 represses and ARPP21 increases GFP reporter expression; co-transfection leads to intermediate expression. Data expressed as mean ± s.d., ***$p < 0.001$, Student's *t*-test, $n = 3$. **f** Representative immunoblot analysis of endogenous PHF6 and MSK1 protein levels in 293T cells. ARPP21 transfection leads to increased protein expression of MSK1 and PHF6. miR-128 mimic transfection results in reduced expression of MSK1 and PHF6 protein. Co-transfection leads to intermediate levels. **g**, **h** Quantification of PHF6 and MSK1 protein expression from five independent experiments as in **f**. *$p < 0.05$, ***$p < 0.001$, one-sample *t*-test against 100%, data represents mean ± s.d. **i** miR-128 overexpression by lentivirus in primary cortical neurons at DIV7 reduces MSK1 and PHF6 protein expression. The arrowhead marks the specific PHF6 band. **j** Quantification of miR-128 overexpression effect on MSK1 and PHF6 protein levels. One-sample *t*-test against 100%, $n = 4$ biological replicates, *$p < 0.05$; **$p < 0.01$. Data expressed as mean ± s.d. Protein levels of ARPP21, MSK1, and PHF6 upon ARPP21 overexpression (**k**) and knockdown (**l**) in primary cortical neurons at DIV7. Quantification of protein expression upon overexpression (**m**) or knockdown (**n**) of ARPP21. One-sample *t*-test against 100%, $n = 6$ biological replicates, *$p < 0.05$. Data expressed as mean ± s.d. Full western blot images are presented in Supplementary Fig. 21

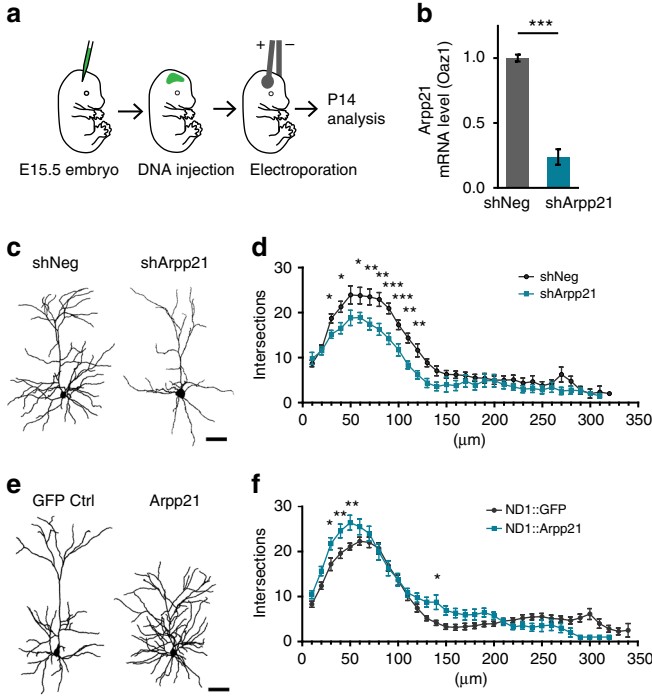

**Fig. 7** Phenotype of ARPP21 loss- and gain-of-function during cortical neuron differentiation in vivo. **a** IUE at E15.5 with *Arpp21* shRNA or overexpression constructs followed by morphological analysis at P14. **b** qRT-PCR of endogenous *Arpp21* mRNA levels in N2A cells confirms efficacy of *Arpp21* shRNA knockdown using *Oaz1* mRNA as standard. ***$p < 0.001$, Student's $t$-test, $n = 3$. Data represent mean ± s.d. **c** Representative reconstructions of a neuron expressing a non-targeting control shRNA (shNeg) and an ARPP21 knockdown (shArpp21) neuron. Scale bar: 50 μm. **d** Sholl analysis of control (shNeg; black) and ARPP21 knockdown (shArpp21; turquoise)-expressing neurons shows significantly decreased dendritic complexity upon ARPP21 knockdown. **e** Reconstructions of representative neurons expressing either GFP or ARPP21 under control of the NeuroD1 (ND1) promoter (plasmids ND1::IRES-GFP or ND1::ARPP21-IRES-GFP, respectively). **f** Sholl analysis reveals a significantly increased dendritic complexity of ARPP21-overexpressing (ND1::Arpp21; turquoise) neurons compared to control neurons (ND1::GFP; black). *$p < 0.05$, **$p < 0.01$, ***$p < 0.001$, two-way ANOVA with Bonferroni post-test. shNeg: $n = 14$ cells. shArpp21: $n = 11$ cells. ND1-GFP: $n = 20$ cells. ND1-Arpp21: $n = 14$ cells. Data expressed as mean ± s.e.m. Scale bar: 50 μm

elevated PHF6 and MSK1 protein levels (Supplementary Fig. 11a, b). Steady-state mRNA levels were also inversely affected by ARPP21 and miR-128, although the effects were considerably smaller compared to the changes at the protein level (Supplementary Fig. 11c-e). To assess if *Msk1* and *Phf6* regulation by ARPP21 might also occur during neurogenesis we first compared public RNA-Seq datasets from TREx-293 cells and cortical projection neurons. This analysis confirmed that the predominant transcript isoforms, including 3′UTR sequences, present in neurons assayed between E16 and postnatal day 1 (P1) are analogous to those present in TREx-293 cells and therefore share the conserved miR-128-binding sites and the uridine-rich sequences recognized by ARPP21 (Supplementary Fig. 12a-c). The abundance of the *Phf6* and the *Msk1* transcripts declines between E16 and P1, consistent with developmental targeting of the miR-128 binding sites present in their 3′UTRs. To confirm interaction of ARPP21 with these transcripts in the embryonic brain we performed RIP of endogenous ARPP21 protein using cortical extracts from E17. Both the *Phf6* and *Msk1* transcripts were significantly enriched

in the immunoprecipitates compared to the control transcripts *Rpl27* or *Oaz1* (Supplementary Fig. 13a-c).

To analyze Arpp21 function in neurons we used lentiviral-mediated knockdown of *Arpp21* and overexpression of miR-128 and ARPP21 in primary cortical neuron cell cultures. As expected, miR-128 significantly reduced both the protein and mRNA levels of neuronal *Phf6 and Msk1* mRNAs (Fig. 6i, j and Supplementary Fig. 14a-c). PHF6 and MSK1 protein levels responded inversely to ectopic ARPP21 and *Arpp21* knockdown (Fig. 6k, l), although the magnitude of the effects was less than in cell lines and was not significant in the case of MSK1 in knockdown cells (Fig. 6m, n). The mRNA response for the two targets was not statistically significant (Supplementary Fig. 14d-f), consistent with the stronger influence of ARPP21 on protein versus mRNA levels observed in cell lines.

**ARPP21 opposes miR-128 functions in dendritic growth in vivo.** Next, we wanted to test if the molecular antagonism between ARPP21 and miR-128 might be relevant to known functions of miR-128 in vivo. Overexpression of miR-128 in upper-layer cortical neurons during embryonic development leads to a significant reduction in the dendritic arborization of the affected neurons, an effect that is largely mediated by PHF6[6]. We therefore performed gain- and loss-of-function experiments for ARPP21 by in utero electroporation (IUE) at embryonic day E15.5 and determined the effect on dendritic arbor complexity by Sholl analysis at P14 (see Fig. 7a for schematic).

Loss-of-function experiments employed a short hairpin construct against transcripts encoding full-length ARPP21 (pshARPP21-GFP) that was validated for efficacy against endogenous *Arpp21* mRNA (Fig. 7b). After IUE the affected neurons were identified by enhanced GFP (eGFP) expression driven by pshARPP21-GFP or a control plasmid expressing a scrambled shRNA (pshNeg-GFP). Comparison of knockdown and control electroporations did not reveal obvious effects on proliferation, neurogenesis, or migration, allowing morphometric analysis of eGFP-expressing neurons at their appropriate developmental positions in the upper cortical layers. eGFP-expressing neurons were patched and filled with biocytin for staining and reconstruction. Representative neurons used for the Sholl analysis are shown in Fig. 7c, e, the full set in Supplementary Fig. 15a-d. The analysis revealed significant reductions at all data points between 30 and 120 μm from the cell bodies (Fig. 7d). This reflects reduced branching of proximal and intermediate dendrites and an overall reduction in dendritic arbor complexity. This is consistent with the general morphology of the neurons, with less-pronounced differences in the length and appearance of apical dendrites compared to basal dendrites. Dendritic morphology could be substantially rescued by co-electroporation of an shRNA-resistant ARPP21 cDNA expression construct, even though the NeuroD1 promoter used to express *Arpp21* initiates expression later in neuronal differentiation than the constitutive promoter used for the knockdown (Supplementary Fig. 16a-d, for complete set of neurons see Supplementary Fig. 17). This result argues against a significant contribution of off-target effects or RISC saturation to the knockdown phenotype.

For gain-of-function experiments ARPP21 was expressed together with eGFP using an IRES construct driven by the neuron-specific NeuroD1 promoter (pND1-ARPP21-IRES-GFP). Compared to the negative control (pND1-IRES-GFP), ectopic ARPP21 had the opposite effect on dendritic complexity as ARPP21 knockdown (Fig. 7e, f). The resulting neurons appeared to be more compact and ramified, with significantly increased branching near the soma (from 30 to 50 μm) and again at a distance of 140 μm. Importantly, co-expression of miR-128

reversed the effects of ARPP21 (Supplementary Fig. 18a-c, complete set of neurons in Supplementary Fig. 19 and 20), indicating that the two exert their effects on dendrite morphogenesis via common pathways. Together, the results of these experiments suggest that cortical dendritic arbors are highly sensitive to ARPP21 dosage during the late embryonic and postnatal stages of growth, and suggest that miR-128 is a negative and ARPP21 a positive regulator of this process.

## Discussion

A variety of regulatory interactions between miRNAs and RBPs are known to occur within the 3′UTR sequences of mRNAs. RBPs can cooperate with miRNAs by facilitating miRNA binding to target transcripts or antagonize miRNA silencing either directly by steric inhibition of miRNA binding[35,36] or indirectly by combatting miRNA effects on translation or mRNA degradation. These cooperative and antagonistic interactions increase the flexibility and complexity of post-transcriptional gene regulation. Here we present a novel, genetically hard-wired regulatory circuit in which ARPP21 can modulate the targeting outcome of its intronic miRNA. ARPP21 binds and stimulates the expression of a subset of miR-128 target mRNAs (Fig. 8a).

Our finding that ARPP21 preferentially binds 3′UTRs suggests that it antagonizes miRNA function in general and miR-128 targeting in particular. However, this does not necessarily imply direct interference with miRNA binding, as we did not observe a statistically significant bias for ARPP21-binding in the vicinity of miR-128 seed sequences (Supplementary Fig. 7d). Furthermore, we were able to show that ARPP21 physically interacts with the eIF4F components eIF4A and eIF4G, and that activation by ARPP21 in a tethering assay is dependent on eIF4G. This suggests that the eIF4F complex may represent a common point of action for ARPP21 and miRNAs, which are thought to suppress eIF4F in the process of miRNA-mediated silencing[38,39]. This regulatory effect of ARPP21 is consistent with the known role of its *Drosophila* ortholog Encore, a positive regulator of gurken mRNA translation[37].

Several intronic miRNAs have been shown to reinforce the regulatory functions of their host proteins. It was therefore surprising that ARPP21 does the opposite and antagonizes the inhibitory effect of miR-128 on several functionally important targets of miR-128, such as *Phf6* and *Upf1*. Since ARPP21 and miR-128 are derived from a single transcriptional unit, this raises the question of how the balance between the inhibitory functions of miR-128 and the activating functions of ARPP21 is regulated. Possibilities include the suppression of miR-128-2 precursor processing in neurogenic progenitors, which might represent a timing mechanism to delay miR-128 accumulation relative to the ARPP21 protein[6]. A negative feedback loop caused by the ability of miR-128 to suppress the ARPP21 mRNA might subsequently promote a rapid switch between the two activities. Methylation and phosphorylation of ARPP21[27,40] may also play a role, for example, via 14-3-3 class proteins, which we show to be phosphorylation-dependent ARPP21 interactors. Each of these mechanisms would increase the plasticity and dynamic range of post-transcriptional control (Fig. 8b).

To test our model of functional antagonism between miR-128 and ARPP21 in development we chose the intellectual disability gene *Phf6* as a highly ranked ARPP21 target in the iCLIP experiment. We previously showed that *Phf6* expression is inhibited by miR-128 during corticogenesis and that PHF6 can rescue the miR-128 overexpression phenotype of reduced dendritic complexity in upper-layer neurons[6]. After showing that ARPP21 enhances *Phf6* expression in cell lines and primary neurons, we confirmed the prediction of our model that ARPP21

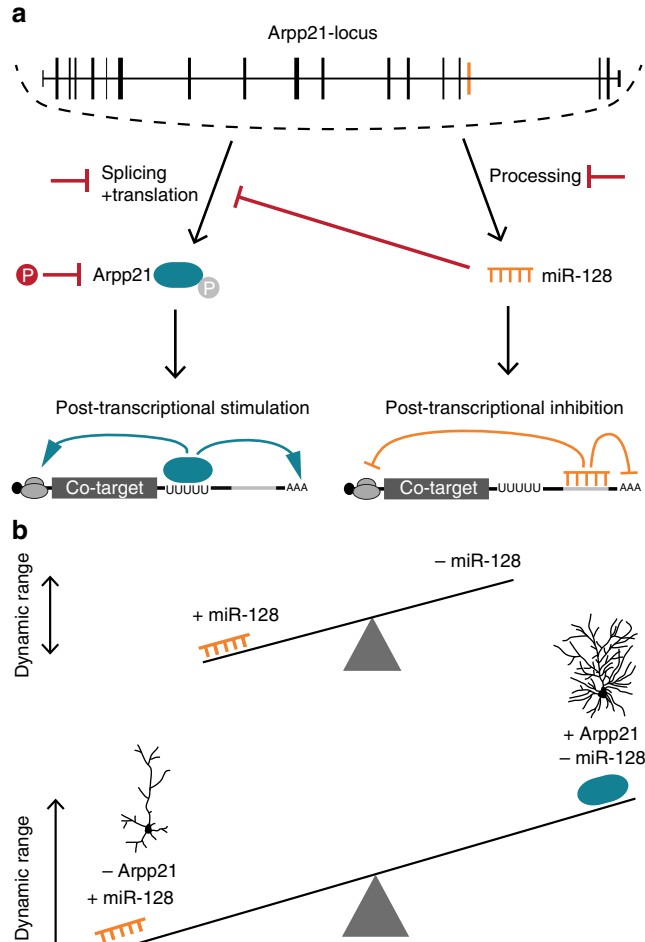

**Fig. 8** Schematic depiction of our model for the miR-128-ARPP21 regulatory circuit and its impact on the post-transcriptional regulation of common mRNA targets. **a** ARPP21 and miR-128 are co-expressed from the same genetic locus. pri-miR-128-2 is generated upon *Arpp21* transcription and subsequently processed by Drosha and Dicer into mature miR-128. miR-128-loaded miRISC binds and downregulates target mRNAs that possess the miR-128 seed match sequence. *Arpp21* is transcribed, spliced, and translated into protein. ARPP21 binds a subset of miR-128 target mRNAs via a uridine-rich sequence motif leading to increased protein expression. Mechanisms that could potentially regulate the relative activities of ARPP21 and miR-128 are marked in red. Alternative splicing of the *Arpp21* mRNA and phosphorylation of ARPP21 protein could regulate its activity. Further, miR-128 can inhibit *Arpp21* expression through a conserved seed match in the *Arpp21* 3′UTR. On the other hand, miR-128 processing is regulated during development. **b** Co-regulated transcripts have a greater dynamic range of gene expression compared to transcripts under control of either miR-128 or ARPP21 acting on their own. Dendritic complexity of cortical neurons is highly sensitive to the relative levels of miR-128 and ARPP21

knockdown should reduce and ectopic ARPP21 increase dendritic complexity. Although increased miR-128 expression could block the effect of ectopic ARPP21 on dendrites, it seems unlikely that *Phf6* is solely responsible for these in vivo effects. However, it likely represents a functionally important target within a network of co-regulated mRNAs that are sensitive to the balance between miR-128 and ARPP21 activity. The functional overlap of miR-128 and ARPP21 might not be restricted to dendritic morphogenesis. For example, the NMD and MAP-kinase signaling pathways are known to be inhibited by miR-128[11,13] and are also enriched for ARPP21-bound mRNAs.

miR-128 is one of the most abundant miRNAs in cortical neurons[41], and is one of the few mammalian miRNAs with a lethal phenotype upon deletion[13]. Our results reveal that miR-128 is not only physically but also functionally embedded in a previously unrecognized post-transcriptional regulatory circuit involving its host genes. The extensive degree of co-regulation we report between miR-128 and ARPP21 is likely to be important for the physiological roles of miR-128 in cortical development and neuronal excitability. The binding repertoire of ARPP21 we observed, however, is by no means restricted to downstream targets of miR-128. Our evidence that ARPP21 acts as a positive post-transcriptional regulator, most likely at the level of translational initiation, is a new example of the complex interactions between miRNAs and RBPs.

## Methods

**Animals**. Experiments were performed with NMRI mice purchased from Charles River, Cologne, Germany. Animals were handled following the rules and regulations of the Berlin Landesamt für Gesundheit und Soziales and the animal welfare committee of the Charité Berlin, Germany.

**Protein isolation from animal tissue**. Tissue was homogenized in ice-cold RIPA buffer (1% NP-40, 0.1% Na-Desoxycholate, 0.1% SDS, 150 mM NaCl, 20 mM HEPES (pH 7.9), 2 mM EDTA, 50 mM NaF, 0.2 mM $Na_3VO_4$, 1 mM dithiothreitol, and 1× protease inhibitor cocktail) by pushing through a 0.9 mm cannula. After 30 s of sonication, samples where centrifuged for 20 min at $14,000 \times g$ and 4 °C. Supernatants were frozen at −80 °C until use.

**SG induction**. To induce SGs, cells were incubated for 45 min in Dulbecco's modified Eagle medium (DMEM), 10% fetal bovine serum (FBS), primary cortical neurons in Neurobasal medium, including supplements, with either 500 mM arsenite or 20 mM clotrimazole, at 5% $CO_2$ and 95% humidity at 37 °C. Stress induction by heat shock was performed at 42 °C with 5% $CO_2$ and 95% humidity. After treatment, cells were washed once with phosphate-buffered saline and fixed for 10 min in 4% paraformaldehyde.

**MS2 tethering assays**. N-terminal FLAG-MS2 fusion constructs were reverse transfected using Lipofectamine2000 (ThermoScientific) in 293T cells together with a dsRed reporter and a GFP reporter construct containing four consecutive MS2 stem-loops in its 3′UTR. Forty-eight hours post transfection the mean GFP fluorescence intensity of the transfected cells (dsRed+) was assessed by flow cytometry. Detailed information on fusion construct cloning can be found under Supplementary Methods.

**ARPP21 expressing doxycycline-inducible TREx cell line**. Flp-In-TREx-293 cells from ThermoScientific were used to generate cell lines expressing murine full-length ARPP21 upon doxycycline addition (iARPP21 cells). ARPP21 was cloned in the pENTR4 vector and transferred by Gateway cloning into the pFRT-FLAG-HA-DEST vector to provide N-terminal FLAG and HA tags, and flanking FRT sites necessary for Flp recombination. Cloning and cell line generation followed established protocols[42].

**Individual-nucleotide resolution crosslinking and immunoprecipitation**. iARPP21 cells were seeded in DMEM with 10% FBS and 1 μg/ml doxycycline. Forty-eight hours after seeding, the cells were subjected to UV-crosslinking and harvested. iCLIP was performed with the mouse monoclonal HA antibody #26183, ThermoScientific. After IP, RNA–protein complexes were washed with high-salt buffer (50 mM Tris–HCl, pH 7.4, 1 M NaCl, 1 mM EDTA, 1% Igepal CA-630, 0.1% SDS, and 0.5% sodium deoxycholate) as described[43]. A detailed protocol with all modifications of the original protocol[43] and the detailed description of the bioinformatic analysis can be found under Supplementary Methods.

**3′UTR targeting assays**. 3′UTRs of ARPP21 and miR-128 target genes were cloned into a modified peGFP-C1 backbone, carrying an in-frame stop codon before the multiple cloning site[44]. Primer sequences are available in the Supplementary Methods. 293T cells were reverse transfected with empty intron-RED vectors with or without intron-RED-miR-128-2[6] and p3xFLAG-CMV-7.1 vectors with or without murine ARPP21. Mean GFP fluorescence of transfected dsRed+ cells was quantified by flow cytometry 48 h after transfection.

**Primary neuron preparation and lentiviral infection**. Primary cortical neurons were prepared from E16.5 wild-type NMRI mice. Cortices were manually dissected and collected in Hanks' balanced salt solution. The tissue was digested using trypsin and treated briefly with DNase. Cells were mechanically dissociated and plated in complete Neurobasal medium (1% P/S, 1% Glutamax, 2% B27, and 25 μM β-mercaptoethanol) in poly-L-lysine-coated (0.1 mg/ml) 12-well or 24-well plates on glass coverslips. For lentiviral infection, cortical neurons were incubated with lentivirus expressing knockdown or overexpression constructs at days in vitro 1 (DIV1). Cells were harvested for RNA and protein analysis at DIV7.

**In utero electroporation and morphometric analysis**. In utero electroporation was performed 15.5 days post gestation following[45]. Information on cloning of the ARPP21 shRNA and overexpression plasmids can be found under Supplementary Methods. At P14 electroporated animals were sacrificed and 300 μm coronal sections were prepared. GFP-positive cells were patched under visual guidance, filled with biocytin, and stained with Alexa-647-conjugated avidin as described previously[6]. Neurons were imaged on an Olympus FluoView confocal microscope with a ×30 silicon immersion objective (numerical aperture 1.1) and reconstructed with ImageJ's Simple Neurite Tracer plugin. Sholl analysis was performed on z-stack maximum intensity projections of the three-dimensional reconstructions with an initial radius = 10 μm, radius step size = 10 μm, and end radius = 350 μm.

**Data availability**. The authors declare that the data supporting the findings of this study and relevant source data are available within the article and its Supplementary Information. Other data and materials are available from the authors upon request. ARPP21 iCLIP sequencing data and all processed iCLIP data are available at the ArrayExpress data repository under accession code: E-MTAB-5911 at https://www.ebi.ac.uk/arrayexpress/experiments/E-MTAB-5911/.

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

## Acknowledgements

We would like to thank D. Richter for excellent technical support and D. Nguyen for critical discussion of the work. We further thank I. Weber for her help with the in situ hybridizations and M. Landthaler for providing the doxycycline-inducible cell line. We thank B. Brokowski and the Charité Viral Core Facility for rapid production of the lentiviral particles. This study was supported by two German Research Foundation (DFG) grants to F.G. Wulczyn and one to R. Backofen (SPP1738), and by a stipendium awarded to P. Knauff by the Charité—Universitätsmedizin Berlin.

## Author contributions

F.R. and F.G.W. conceived of the project and designed the experiments. D.M. analyzed the iCLIP sequencing data. S.G. performed neuron labeling, microscopy, and neuron reconstruction. P.K. performed miR-128 gain-of-function experiments in neurons, in utero electroporations for the Supplementary Information, and prepared the primary neuronal cultures. M.E. conducted the LC-MS/MS measurement and analysis. F.R. performed all other experiments. I.V., R.B., and F.G.W. supervised the project. F.R. and F.G.W. wrote the manuscript with input from S.G., P.K., I.V., and R.B.

## Additional information

**Competing interests:** The authors declare no competing interests.

