## [Peer Review File(PDF 777 kb) · Nature Communications]

Reviewers' expertise:

Reviewer #1: neuronal miRNA, neuronal morphogenesis;

Reviewer #2: neuronal RNA, RNA-binding proteins;

Reviewer #3: miRNA, translational control.

Reviewers' comments:

Reviewer #1 (Remarks to the Author):

Here are some issues with this manuscript:

1. They should show these data: "We consistently observed perinuclear enriched, granular staining but also occasional cells containing larger cytosolic aggregates that resembled cytosolic stress granules (SGs) (data not shown)."
2. They should use at least one more marker for SGs: "To test if R3HDM1 or ARPP21 can be recruited to SGs, HeLa cells were treated with arsenite and SG formation was monitored by staining for the marker protein eIF3 η (Fig. 2C, D)."
3. This conclusion is not true: "The predicted RNA-binding domains, however, are dispensable, suggesting that protein-protein interactions are responsible for the recruitment of ARPP21 to SGs." The RNA binding domain deleted form of ARPP1 may be interacting with RNAs in the SG and in fact this is likely.
4. This statement needs more explanation and examples of how they draw the sweeping conclusion of increased translation: "The opposite effect was observed with a full-length ARPP21-MS2 fusion, which increased GFP fluorescence about two-fold compared to the MS2-only control (Fig. 3B, C)."
5. This experiment requires additional controls including determination of overlapping protein sets with non-relevant proteins: "LC- MS/MS analysis identified 150 proteins with at least two-fold or higher peptide intensities in both the ARPP21 and R3HDM1 IPs compared to control IPs from cells not expressing either host protein (Supplemental Table S2)."
6. Binding of eIF4G and eIF4A needs to be presented in terms of the stringency with which they determined the association.
7. They say, "Crosslinking followed by IP for ARPP21 at high stringency revealed a distinct autoradiograph signal corresponding to ARPP21-RNA complexes." What is high stringency and what is proof that the band is ARPP21 other than the presence of the HA tag?
8. Unless they can show a response curve these signal strength statements are not valid. For example: "Signal strength was strongest for the full-length protein and was undetectable with a C-terminal construct lacking the R3H and SUZ domains. An R3H-only construct had substantially weaker binding activity than either the full-length protein or a minimal construct containing only the R3H and SUZ domains." In general CLIP is not considered particularly quantitative with the degree of resolution they appear to be claiming.
9. They say, "This revealed a strong preference for single-stranded conformations at the central, crosslinked nucleotide of the motif ..." This result and possibly the uridine preference does not necessarily indicate a preferred binding site for the interaction. It indicates a region that is preferentially crosslinked, which is not identical to the binding

site.

10. They say, "These results prompted us to compare the KEGG pathway for ARPP21 targets with that of 1132 mRNAs predicted to harbor conserved miR-128 binding sites derived from the TargetScan 7.1 database." This approach does not work—there are too many false positives and cryptic sites not found in TargetScan for this method to give reliable results. They need to perform experimental detection miR128 targets and a more complete statistical analysis.

11. This statement requires much stronger statistical support: "This comparison showed that the 3'UTRs of ARPP21 target mRNAs harbor a significantly higher number of miR-128 7mer seed sequences compared to the 3'UTRs of ARPP21 non-targets (Fig. 5C, left panel)."

12. They say: "To independently verify the iCLIP results for these mRNAs RNA-immunoprecipitation (RIP) was performed using 293T cells transiently expressing FLAG-tagged ARPP21 (Supplemental Fig. S6A). " RIP is very misleading and as presented here has insufficient controls.

Reviewer #2 (Remarks to the Author):

In this well-presented, convincing work, the authors show that the RNA-binding protein Arpp21 expressed by the miR-128 host gene is counteracting the function of miR-128, a miRNA that is important for cortex development and neuronal differentiation. In my opinion, the figures are clear and convincing and the manuscript is well-written. For a revised version of this study, however, it would be critically important to strengthen the link between the work on Arpp21 in cell lines and the final experiment in neurons and in vivo. Therefore, the authors shall include additional experiments using primary neurons, where the protein is endogenously expressed.

Comments and suggestions:

1) How does ARPP21 protein localize in neurons? Does it localize to dendrites (branching)? It would improve the meaning of Fig 2, to see whether the localization to stress granules in HeLa cells can be confirmed in neurons. It would be desirable to incorporate these data better in the rest of the paper. One option would be to use cellular stress in combination with the MS2-reporter tethering assay from Fig 3, to look at the consequences for the function of Arpp21, when it localizes to stress granules.

2) Using a tethering assay and co-IP of overexpressed Arpp21, the authors show that Arpp21 increases MS2 reporter expression, possibly through interaction with the translation initiation factors Eif4g and Eif4a (Fig. 3). However, as Arpp21 is an RNA-binding protein, it is necessary to also show the reporter RNA levels for the tethering assay, even if the known RNA-binding domains of Arpp21 alone did not increase reporter expression.

3) The IP results from N2A cells showing endogenous Eif4 proteins interacting with Arpp21 look promising (Fig 3E-G). It would be interesting to see, whether Arpp21 depleted for the C-terminal domain is not competent for this interaction, as indicated by the tethering assay. A complete reversal of the positive effect on expression of Arpp21 in the tethering assay (Fig 3K) by co-downregulation of both Eif4G and Eif4a would further strengthen the positive translational function (also show RNA levels for this assay). As the authors are exclusively working with cell lines at this point and not showing any data from primary neurons or brain

lysate, polysome profiling or a translation assay should be a straight forward experiment to strengthen the positive translational effect of Arpp21.

4) For the interpretation of the RNA binding domains in Fig 4A-C, I suggest stepping back from the cooperative RNA binding of the two RNA-binding domains based only on the autoradiogram. It looks more like the C-terminal domain seems to be stabilizing the RNA interaction of both RNA-binding domains. Is this experiment really important for the manuscript as it is not followed up subsequently, e.g. for the iCLIP experiments.

5) In order to identify the mRNA targets and binding specificity of Arpp21, the authors successfully did iCLIP from an inducible cell line. The data are convincing and the overlap between miR-128 and Arpp21 targets is really exciting (Fig 4-5). However, one interesting aspect could be further investigated: Are the iCLIP sites/Arpp21 binding sites in proximity to miR-128 seeds, or is there a random distribution? This analysis could give an insight, whether Arpp21 and miR-128 might also compete for binding of the 3'-UTR on top of their opposing function.

6) In Fig 5, the authors use a GFP assay and Western blot from 293T cells to show the molecular opposing effect of Arpp21 and mir-128 overexpression on common target 3'-UTRs found in the iCLIP dataset. The results are surprisingly strong, especially for miR-128. Again, one would like to see the RNA levels in comparison. In addition, it would be interesting, which effect the different Arpp21 mutations (RNA-binding domain deletion, C-terminal deletion) have on the 3'-UTR in the GFP-assay. In the context of the title (focusing on the opposing function of the miR-128 and Arpp21), it would be important to see how the target mRNAs (or proteins) are behaving in primary neurons upon Arpp21 and miR-128 up- or downregulation. This would also improve the link to the in vivo data from Fig 6 and further boost the significance of this manuscript.

7) In Fig 6, the authors should include a rescue experiment by co-expression of a miR-128 sponge or miR-128 mimic, with the shArpp21 or Arpp21 respectively, thereby showing the opposing function in vivo.

8) In Fig 7, the authors could include the negative feedback loop from mature miR-128 to the Arpp21 mRNA, since miR-128 is binding and downregulating the long Arpp21 isoform (see Fig S1A). Further, I would suggest removing the terms stability and decay from the chart, since no data shown are supporting any of these conclusions.

Minor comments:

1) Check writing of the R3hdm1/2 genes or proteins in the legend to Fig 1.

2) Check first sentence in legend of Fig 2, there is no "host protein" for a micro-RNA. I would suggest, removing the link to the miR-128, since you are only showing data from Arpp21 and R3hdm. The authors showed in Fig 1 that those are the host genes.

3) There is an minor inconsistency in the data analysis in Fig 5H/I and Fig 5K/L, the analysis of the significance is different.

Reviewer #3 (Remarks to the Author):

Here, Rehfeld and colleagues investigate the relationship between a gene that codes for the Arpp21 protein, and also codes for the intronic miR-128 microRNA. They provide evidence

that Arpp21 post-transcriptionally enhances gene expression and binds to a subset of bioinformatically-predicted miR-128-targeted mRNAs. Overall, their model is that Arpp21 antagonizes miR-128 silencing capacity on select mRNAs. Moreover, they posit that this molecular mechanism regulates dendritogenesis in terminally differentiated cortical neurons in the mouse brain. Overall, this is a very interesting manuscript and the data are of high quality. That being said, several points need to be addressed:

Major comments:

- 1) While it is quite interesting that Arpp21 enhances gene expression, and Arpp21 interacts to a certain degree with eIF4G and eIF4A, I think that the authors' statement that this interaction is causative for upregulation of gene expression is a bit overreaching. One thing that I think could strengthen this argument is if the authors test whether their Arpp21 deltaC-term mutant, which cannot enhance gene expression of their reporter, is unable to co-precipitate eIF4G and eIF4A. If it can't then this would strengthen their argument. If it doesn't, then maybe eIF4G/eIF4A interaction may not play a major role in the upregulation of gene expression they observe.
- 2) For IUE experiments, they need to use a separate shRNA to validate and make sure not off-target. Alternatively, the authors could co-transfect a Arpp21 rescue construct along with the shRNA to test whether this ameliorates the effect of the shRNA alone.
- 3) When testing various 3'UTRs as being putative miR-128 targets, the authors use a negative control reporter containing five miR-128 target sites that are perfectly complemented to miR-128 but have no poly(U)-rich sequence that can bind Arpp21. Unfortunately, this is not a proper control as this reporter will be cleaved by miR-128 via RNAi. This is due to the fact that miRNAs that are perfectly complementary to target sequences act as siRNAs. The authors need to have a control that is targeted by miR-128 via imperfect complementarity. I would use a bona fide miR-128 targeted 3'UTR to model what kind of sequences they can use for their control reporter.
- 4) A major limitation of the CLIP-Seq data obtained is that it is all carried out in a non-neuronal cell line (i.e. HEK-293 cells), yet the main impact points of the study investigate Arpp21 function in the nervous system. The author should validate a select population of their Arpp21-targeted mRNAs in either primary neuronal cultures, or neuronal cell lines such as N2a or SH-Y5Y. This would help to confirm that their iCLIP datasets do not just occur in the context of HEK-293 cells.
- 5) The authors posit that Arpp21 expression in neurons is critical for regulating dendrite arborisation in part by opposing miR-128 action. Unfortunately, the data support a genetic linkage, but not a direct mechanistic link between miR-128 and Arpp21. Can the authors provide any evidence that Arpp21 is acting on miR-128-targeted mRNAs in neurons?

Reviewer #1 (Remarks to the Author):

Here are some issues with this manuscript:

1. They should show these data: “We consistently observed perinuclear enriched, granular staining but also occasional cells containing larger cytosolic aggregates that resembled cytosolic stress granules (SGs) (data not shown).”

The quoted statement has two parts. The predominant staining pattern was “perinuclear enriched, granular staining” as seen in Fig. 2a and 2b. Although we believe this is a legitimate situation to employ “data not shown”, to comply with the reviewer’s request an example of “occasional cells containing larger cytosolic aggregates that resembled cytosolic stress granules (SGs)” is now included as Supplementary Fig. 2a.

2. They should use at least one more marker for SGs: “To test if R3HDM1 or ARPP21 can be recruited to SGs, HeLa cells were treated with arsenite and SG formation was monitored by staining for the marker protein eIF3 η (Fig. 2C, D).”

We would like to point out that in the original manuscript we showed results with two SG markers, eIF3 η in the main Figure and FXR2, in Supplementary Fig. 2d and 2e. Nevertheless, to comply with the reviewer’s request we have confirmed these results with a third marker for SGs, G3BP, in Supplementary Fig. 2f.

3. This conclusion is not true: “The predicted RNA-binding domains, however, are dispensable, suggesting that protein-protein interactions are responsible for the recruitment of ARPP21 to SGs.” The RNA binding domain deleted form of ARPP1 may be interacting with RNAs in the SG and in fact this is likely.

The reviewer is correct that there is considerable evidence for the ability of so-called unstructured domains to interact with RNA. Our inference that this may not be the case for the C-terminal domain of ARPP21 is based on the lack of detectable RNA binding activity by this domain in the crosslinking experiment (Fig. 4 a-c) and the fact that its transactivation activity on confirmed mRNA targets of ARPP21 is dependent on physical tethering of the domain to mRNAs via MS2 (Fig. 3c,d).

We agree that we have not formally ruled out potential direct C-term-RNA interactions under stress conditions and have edited the text to reflect the reviewer’s concern.

4. This statement needs more explanation and examples of how they draw the sweeping conclusion of increased translation: “The opposite effect was observed with a full-length ARPP21-MS2 fusion, which increased GFP fluorescence about two-fold compared to the MS2-only control (Fig. 3B, C).”

We regret making the impression that we are drawing “sweeping conclusions” about mechanisms. What we consistently observe is evidence for specific interactions between ARPP21 and mRNAs (iCLIP, RIP, reporter assays) and increased expression of the protein products of these same mRNAs (reporter

and tethering assays, transfection assays in cell lines now confirmed in primary neurons - Fig. 6 k-n). At the request of Reviewer 2 (Point 6) we now provide results of mRNA quantifications for each of these transactivation assays. Together with the evidence for physical and functional interactions with the translation initiation factors eIF4A and eIF4G (Fig. 3g-i) the results suggest - we think strongly - but do not prove that ARPP21 acts as a translational activator. Whether ARPP21 directly or indirectly mediates mRNA stability or acts by additional mechanisms remains to be determined, as discussed elsewhere in this response and in the manuscript. Therefore, we carefully edited any text passages in the manuscript that referred to ARPP21 as a translational activator to the more general term post-transcriptional activator.

For the quote in question: we believe the tethering experiment includes all necessary controls and is in principle widely accepted as evidence for post-transcriptional activation or suppression activity. We have carefully edited the text to provide more clarity about the basis of our interpretations and made every attempt to avoid overinterpretation. We thank the reviewer for this comment and the opportunity to improve our writing and presentation.

5. This experiment requires additional controls including determination of overlapping protein sets with non-relevant proteins: "LC- MS/MS analysis identified 150 proteins with at least two-fold or higher peptide intensities in both the ARPP21 and R3HDM1 IPs compared to control IPs from cells not expressing either host protein (Supplementary Table S2)."

In our view, the experimental design of this experiment includes all essential and standard control conditions and should satisfy the reviewer's requirements. The peptide recovery using anti-FLAG antibody and extracts from cells transfected with bait-proteins is quantified and compared to recovery with anti-FLAG precipitates from transfected control cells. Although related, the two bait proteins ARPP21 and R3HDM1 are divergent enough to allow us to test for specificity of the pulldowns, as documented by the specific recovery of prey proteins in the ARPP21 compared to the R3HDM1 IPs. After normalization to the negative control, 150 proteins displayed 2-fold or more enrichment for either or both of the bait proteins. Of these, 31 displayed more than 2-fold specific enrichment for either ARPP21 or R3HDM1. Together, these results indicate that the pulldown is both reproducible (the set of shared interactors) as well as specific (the set of interactors specific for either ARPP21 or R3HDM1).

Because ARPP21 and R3HDM1 share many similarities (cytoplasmic localization, RNA-binding activity, post-transcriptional regulatory ability), we believe this is a better control than comparison to a hypothetical unrelated protein such as a nuclear transcription factor or membrane protein. The primary data will be uploaded and publicly available upon publication so that the kind of comparison the reviewer suggests (unrelated proteins) can be made using public databases that archive the results of many similar experiments with similar conditions.

Importantly, these results gave us sufficient confidence in the specificity of the potential interactions to validate a panel of candidates. Validation included

phosphorylation-dependent interactions with 14-3-3 proteins (Supplementary Fig. 4c) as well as homomeric interactions between R3HDM1 and ARPP21 themselves (Fig.3f and Supplementary Fig. 4b). Note that the phosphorylation dependence of the 14-3-3 interaction is also a test for specificity of the IPs, and the validation of the homomeric interactions include a demonstration that the interaction is seen using a second epitope tag (anti-GFP). We included these validations in the manuscript partly because of their intrinsic interest but also because they are controls for the MS screen and the identification of eIF4A and eIF4G as potential mediators of ARPP21-dependent transactivation. The reviewer is correct that our focus on proteins that interact with both ARPP21 and R3HDM1 increases the danger that we may have chosen non-specific interactors for study. Nevertheless, our rationale for this approach is sound: both proteins act as post-transcriptional activators and this is the biological activity we wish to investigate.

We took care to show that interactions between ARPP21 and eIF4A are not dependent on the use of the anti-FLAG antibody (Fig. 3h,i) or the choice of matrix (Dynabeads or FLAG-conjugated agarose). Most importantly, in response to Reviewer 2 and 3 we now present a deletion analysis to determine the domain in ARPP21 responsible for the eIF4G interaction. We show that a construct containing the dual RNA-binding domains but lacking transactivation activity in the tethering assay does not interact with eIF4G. A construct containing the C-terminal domain that lacks RNA-binding activity in the cross-linking experiment (Fig. 4b,c) but has transactivation activity in the tethering assay (Fig. 3c) does interact with eIF4G (Supplementary Fig. 4e). We hope that this new data will alleviate the reviewer's concerns.

6. Binding of eIF4G and eIF4A needs to be presented in terms of the stringency with which they determined the association.

We thank the reviewer for pointing out this omission and have provided this information in our revised Fig. 3g-i and Supplementary Methods.

7. They say, "Crosslinking followed by IP for ARPP21 at high stringency revealed a distinct autoradiograph signal corresponding to ARPP21-RNA complexes." What is high stringency and what is proof that the band is ARPP21 other than the presence of the HA tag?

As stated in the Supplementary Methods section, high stringency refers to the washing conditions optimized by the Ule lab for these experiments¹: 1M NaCl in the presence of 0.5% Sodium Deoxycholate, 0.1% SDS, 1% NP40). The reviewer is correct, the reader should not have to look for this information and we have replaced this qualitative description with an explicit one to clarify this point.

The evidence that the band is indeed ARPP21 is, due to space constraints, presented in Supplementary Fig. 5b and Fig. 4b and 4c. In Supplementary Fig. 5b we show that the RNA-protein complex resolves to the expected molecular weight of ARPP21 after treatment with RNase (cf Lanes 1 and 3). The band is completely dependent on doxycycline induction of the ARPP21-transgene in the inducible HEK293 cell line we used (Supplementary Fig. 5b,

cf Lanes 1-3 and 4-6). This result was consistent for all six experimental replicates and all pilot experiments. In Figure 4, the same CLIP-procedure after transfection of FLAG-tagged constructs in HEK293 cells and anti-FLAG IP leads to recovery of a complex with comparable mobility (cf Fig. 4b Lanes 1 and 2 with Fig. 4c Lanes 1 and 2), demonstrating that comparable results are obtained with either anti-HA or anti-FLAG affinity purification and that the complex is dependent on ectopic expression of full-length ARPP21 and cannot be specific to the anti-HA IP. Furthermore, no other RNA-protein complexes that might be due to endogenous RNA-binding activity in the HEK293 cells we used were detectable (Supplementary Fig. 5b lanes 4-6 and Fig. 4c Lanes 1-6). We think this evidence is conclusive and have carefully edited the text to address the reviewer's concerns.

8. Unless they can show a response curve these signal strength statements are not valid. For example: "Signal strength was strongest for the full-length protein and was undetectable with a C-terminal construct lacking the R3H and SUZ domains. An R3H-only construct had substantially weaker binding activity than either the full-length protein or a minimal construct containing only the R3H and SUZ domains." In general CLIP is not considered particularly quantitative with the degree of resolution they appear to be claiming.

Although we intended our statement to merely reflect the apparent qualitative differences in signal strength we observed, we agree with the reviewer that the evidence is not sufficient to make quantitative claims and have omitted these comparisons in the revised text.

9. They say, "This revealed a strong preference for single-stranded conformations at the central, crosslinked nucleotide of the motif ..." This result and possibly the uridine preference does not necessarily indicate a preferred binding site for the interaction. It indicates a region that is preferentially crosslinked, which is not identical to the binding site.

All indications from our data set suggest that this is indeed the binding site, or at least part of the binding site; an evaluation we have confirmed in consultation with independent experts. We provide additional support for the functional significance of the polyU motif by showing that deletion of a uridine-rich stretch (72 uridines of 88 consecutive nt) present in the 3'UTR of the mouse *Upf1* mRNA impairs transactivation by ARPP21 in a reporter assay (Supplementary Fig. 9a and 9c). Nevertheless, we agree with the reviewer that we should entertain alternative explanations in the interpretation of this data and have modified the text in the revised version to reflect the reviewer's concern.

10. They say, "These results prompted us to compare the KEGG pathway for ARPP21 targets with that of 1132 mRNAs predicted to harbor conserved miR-128 binding sites derived from the TargetScan 7.1 database." This approach does not work—there are too many false positives and cryptic sites not found in TargetScan for this method to give reliable results. They need to perform

experimental detection miR128 targets and a more complete statistical analysis.

The reviewer is correct that the TargetScan predictions are subject to false positive and false negative results (some predictions may be false and true targets may be missed) and that this limits the sensitivity of our analysis. Of course, the same is true for any other experimental approach. However, these limitations in the use of TargetScan should lead to false negatives in the KEGG analysis but are unlikely to result in false positives. In point of fact the analysis allowed us to make and test predictions of mRNAs that may be co-regulated by miR-128 and ARPP21.

We do not claim that the KEGG analysis is a complete description of the biological activity of miR-128. We do point out in the text that experimental detection of miR-128 targets support the validity of the KEGG analysis. In other words, we are using this analysis to confirm that experimentally determined targets of miR-128 are enriched in the mRNAs we identified as substrates for ARPP21 in our iCLIP experiment. Schaefer et al. performed AGO2-CLIP experiments and TRAP analysis in control and miR-128 deficient brains². They also identified and validated the MAP-kinase/neurotrophin signaling pathways as significantly misregulated in miR-128 deficient brains. The functional categories obtained by performing a KEGG enrichment analysis of the putative direct targets they identified is in good general agreement with our analysis based on TargetScan. For example, the MAP-kinase signaling pathway is significantly enriched in the AGO2-CLIP sequences that contain miR-128 seeds (our unpublished observations using their data set). The same is true for an additional set of putative direct targets obtained in the same study identified by TRAP: strong enrichment for MAP-kinase and neurotrophin signaling.

In a similar vein, the mRNA surveillance pathway was identified in an experimental screen for miR-128 targets performed in undifferentiated stem cells³. The same paper and a subsequent study provide strong evidence for the relevance of miR-128 mediated regulation of this pathway⁴.

For these reasons, we do not agree with the reviewer's contention that our approach "doesn't work", in fact our analysis is in good agreement with a substantial body of published experimental results. We believe this is sufficient justification for using this approach. At the reviewer's request, we have revised the presentation of Fig. 5a and now provide a statistical analysis. The overlap we observe between predicted miR-128 targets and ARPP21 targets is highly significant ($p < 4.7 \times 10^{-41}$).

11. This statement requires much stronger statistical support: "This comparison showed that the 3'UTRs of ARPP21 target mRNAs harbor a significantly higher number of miR-128 7mer seed sequences compared to the 3'UTRs of ARPP21 non-targets (Fig. 5C, left panel)."

At the reviewer's request we now provide statistical support, this analysis is provided in the revised version of the Fig. 5c. The calculated p-value for miR-128 seed enrichment in ARPP21 targets is $< 2.2 \times 10^{-16}$.

12. They say: "To independently verify the iCLIP results for these mRNAs RNA-immunoprecipitation (RIP) was performed using 293T cells transiently expressing FLAG-tagged ARPP21 (Supplemental Fig. S6A). "RIP is very misleading and as presented here has insufficient controls.

Taken by itself, we would agree that RIP can be misleading and this is why this experiment is included as a Supplementary Figure. However, we believe the data are important and valid as one link in a chain of evidence and believe that in this context RIP is a common and valuable test of iCLIP results. At a technical level, for iCLIP we used an inducible cell line and HA-tagged ARPP21 for affinity purification. For RIP we use transfected cells and FLAG-tagged ARPP21. The reviewer is correct that iCLIP is the superior method, however, one possible criticism is that iCLIP may be biased toward low affinity binding events on highly expressed targets. For this reason, we feel that the degree of enrichment we see in the RIP for our panel of test RNAs (up to 35-fold for *Phf6*) is a valid and important control. Furthermore, there is good qualitative agreement between the ranking of the target genes in the iCLIP experiment and the degree of enrichment after RIP. We would point out that we normalize the input recovery to *Rpl27* for this calculation, but include two additional negative control RNAs that were not identified in the iCLIP experiments (*Gapdh* and *18S* RNA).

For this experiment we relied on a data analysis mode used by the Landthaler group in a recent publication because we believe it is valid and intuitive and also controls for issues of non-specific binding by comparing the efficiency of recovery among multiple transcripts⁵. To satisfy the reviewer's concerns about the controls involved, we have included additional information in Supplementary Fig. 7b. We now display the percentage of input recovery of the ARPP21 and the control IP for each transcript.

Reviewer #2 (Remarks to the Author):

In this well-presented, convincing work, the authors show that the RNA-binding protein Arpp21 expressed by the miR-128 host gene is counteracting the function of miR-128, a miRNA that is important for cortex development and neuronal differentiation. In my opinion, the figures are clear and convincing and the manuscript is well-written. For a revised version of this study, however, it would be critically important to strengthen the link between the work on Arpp21 in cell lines and the final experiment in neurons and in vivo. Therefore, the authors shall include additional experiments using primary neurons, where the protein is endogenously expressed.

We thank the reviewer for this quite positive general assessment and have directed our efforts toward substantiating the relevance of our findings for neuronal systems, as suggested. We hope to convince the Reviewers and the Editors that we have made substantial progress toward meeting these essential criteria.

Comments and suggestions:

1) How does ARPP21 protein localize in neurons? Does it localize to dendrites (branching)? It would improve the meaning of Fig 2, to see whether the localization to stress granules in HeLa cells can be confirmed in neurons. It would be desirable to incorporate these data better in the rest of the paper. One option would be to use cellular stress in combination with the MS2-reporter tethering assay from Fig 3, to look at the consequences for the function of Arpp21, when it localizes to stress granules.

The reviewer raises an interesting point: what happens during cellular stress when ARPP21 enters stress granules. However, this is difficult to address with the MS2 system because we would not be able to easily distinguish post-stress reporter expression from the background of GFP expression that occurs prior to stress induction. The alternative of prolonged stress during the 48 hours between transfection and measurement would of course kill the cells. We have therefore concentrated on the other suggestions in the comment.

We present confocal images of endogenous ARPP21 staining in primary cortical neuronal cultures (revised Fig. 2k and revised Supplementary Fig. 3). These results indicate that ARPP21 is primarily found in the neuronal cell bodies. We also observe what appear to be dendritic granules, primarily in proximal dendritic segments. However, at this time we cannot rule out significant functions for ARPP21 in either synapses or axons, particularly in more mature neuronal networks. Neither ARPP21 or R3HDM1 have, to our knowledge, been detected in the many proteomic studies of synaptic proteins.

To strengthen the connection to neuronal functions we have treated these same neuronal cultures with arsenite to induce stress granules. The results confirm our previous experiments in transfected cell lines. Endogenous neuronal ARPP21 substantially co-localizes with the stress granule marker eIF3 η (revised Fig. 2l).

2) Using a tethering assay and co-IP of overexpressed Arpp21, the authors show that Arpp21 increases MS2 reporter expression, possibly through interaction with the translation initiation factors Eif4g and Eif4a (Fig. 3). However, as Arpp21 is an RNA-binding protein, it is necessary to also show the reporter RNA levels for the tethering assay, even if the known RNA-binding domains of Arpp21 alone did not increase reporter expression.

At the reviewer's request, these data are now included in the Supplement (Supplementary Fig. 4g). It is notoriously difficult to distinguish between direct and indirect effects on translational efficiency vis-à-vis mRNA stability due to their mechanistic connections. It is well established that polyA binding proteins (PABPs) promote translational initiation by interacting with the CAP-binding eIF4F complex while simultaneously suppressing CCR4-NOT deadenylation activity. We feel it is beyond the scope of this study to rigorously distinguish between direct and indirect effects on mRNA stability and we scrupulously avoid making such claims. It would not be surprising if ARPP21 also interacts with the mRNA degradation machinery as well as

translational initiation factors. However, our protein-protein interaction data point to direct interactions with initiation factors.

In general, we find that ARPP21-mediated transactivation is accompanied by an increase in steady-state mRNA levels. With the exception of the tethering assay, the magnitude of the response at the protein level is greater than at the mRNA level (see also response to Point 6).

Interestingly, the effects of ARPP21 at the protein and mRNA levels could be uncoupled: knockdown of the translational initiation factor eIF4G in the tethering assay suppresses the effect of MS2-ARPP21 on reporter expression (revised Fig. 3k). In contrast, the positive effect of tethered ARPP21 on steady-state mRNA levels was unaffected (Supplementary Fig. 4f,g). We believe that the eIF4G-dependence supports a role for ARPP21 in promoting initiation complex formation or stability. We caution against over-interpreting the effects of tethered ARPP21 on steady-state mRNA levels, since this may be a non-physiological result that is due to the exceptionally strong binding of the tethered proteins via the multiplexed MS2 binding sites.

3) The IP results from N2A cells showing endogenous Eif4 proteins interacting with Arpp21 look promising (Fig 3E-G). It would be interesting to see, whether Arpp21 depleted for the C-terminal domain is not competent for this interaction, as indicated by the tethering assay. A complete reversal of the positive effect on expression of Arpp21 in the tethering assay (Fig 3K) by co-downregulation of both Eif4G and Eif4a would further strengthen the positive translational function (also show RNA levels for this assay). As the authors are exclusively working with cell lines at this point and not showing any data from primary neurons or brain lysate, polysome profiling or a translation assay should be a straight forward experiment to strengthen the positive translational effect of Arpp21.

In response to these comments we have now performed this important test of our model and show that the C-terminal domain is necessary and sufficient for the eIF4G and eIF4A interaction by co-immunoprecipitation (Supplementary Fig. 4e). A construct containing the two RNA-binding domains but lacking the C-terminal domain is unable to retrieve endogenous eIF4G and eIF4A. As a control, in the same IP we show the inverse result with respect to the interaction with 14-3-3-ε. We think this is a nice demonstration of the specificity of the co-IPs. These new results are consistent with our model for ARPP21 and we thank the reviewer for the suggestion.

We have tried to knockdown eIF4A in order to comply with the reviewer's second request using three individual siRNA molecules. Although one of the siRNAs corresponds to a published sequence (siEif4a1#7)⁶ in our hands we were unable to achieve even 50% knockdown efficiency (Reviewer Response Fig. 1). This is most likely due to the high abundance and long half-life reported for eIF4A⁷.

Reviewer Response Figure 1

Our laboratory specializes in neurodevelopment and our primary focus is on *in vivo* functions for miR-128 and ARPP21. We have therefore concentrated on answering the criticisms in Points 1 and 7. The establishment of an *in vitro*, semi-purified translational assay is outside our core expertise. We hope to initiate a collaboration to study the molecular details of ARPP21 activity in the future. We also hope that our efforts toward elucidating the molecular functions for ARPP21 will be seen as an important contribution for future mechanistic as well as developmental studies. We ask the reviewer's agreement that further detailed mechanistic experimentation is beyond the scope of this study given our focus on neurobiological aspects.

4) For the interpretation of the RNA binding domains in Fig 4A-C, I suggest stepping back from the cooperative RNA binding of the two RNA-binding domains based only on the autoradiogram. It looks more like the C-terminal domain seems to be stabilizing the RNA interaction of both RNA-binding domains. Is this experiment really important for the manuscript as it is not followed up subsequently, e.g. for the iCLIP experiments.

We have rewritten this section to curb our overenthusiasm in the initial submission, also in response to a similar criticism made by Reviewer 1. We agree that this experiment should be interpreted more carefully and refrain from any quantitative statements in the revised version. We have elected to keep the data in the manuscript because it is an important independent test of the iCLIP conditions, as requested by Reviewer 1. We also hope that this deletion analysis can serve as a foundation for future structural work on the ARPP21-RNA interaction. For this reason, the results may be valuable for our readers.

5) In order to identify the mRNA targets and binding specificity of Arpp21, the authors successfully did iCLIP from an inducible cell line. The data are convincing and the overlap between miR-128 and Arpp21 targets is really exciting (Fig 4-5). However, one interesting aspect could be further investigated: Are the iCLIP sites/Arpp21 binding sites in proximity to miR-128 seeds, or is there a random distribution? This analysis could give an insight, whether Arpp21 and miR-128 might also compete for binding of the 3'-UTR on top of their opposing function.

We thank the reviewer for this comment and note that it is directly relevant as a possible rebuttal to one of the criticisms of Reviewer 1.

The reviewer raises a good point regarding spatial relations between ARPP21 and miR-128 binding sites. We now present a statistical analysis to determine

the average proximity of ARPP21 binding sites vs actual miR-128 seed sequences or a randomly distributed sequence. This shows a weak preference for proximity that does not reach statistical significance (Supplementary Fig. 7d). This result argues against direct steric competition with miRNA binding or miRISC engagement as a mechanism for the majority of co-regulated mRNAs, although it does not rule it out for individual mRNAs. This result is also consistent with one possible interpretation of our work. To simplify a complex field, current models for miRNA function suggest the miRISC complex recruits deadenylation factors to enhance mRNA decay. Furthermore, the miRISC may directly interfere with translational initiation, for example by interacting with the alternative, inhibitory CAP binding protein eIF4E2/4E-HB according to recent studies⁸. In this regard, ARPP21 might be a direct molecular antagonist of miRNAs by interacting with eIF4G and eIF4A1 to promote initiation. If true, this mechanism should not be dependent on proximity to miRNA binding sites.

6) In Fig 5, the authors use a GFP assay and Western blot from 293T cells to show the molecular opposing effect of Arpp21 and mir-128 overexpression on common target 3'-UTRs found in the iCLIP dataset. The results are surprisingly strong, especially for miR-128. Again, one would like to see the RNA levels in comparison. In addition, it would be interesting, which effect the different Arpp21 mutations (RNA-binding domain deletion, C-terminal deletion) have on the 3'-UTR in the GFP-assay. In the context of the title (focusing on the opposing function of the miR-128 and Arpp21), it would be important to see how the target mRNAs (or proteins) are behaving in primary neurons upon Arpp21 and miR-128 up- or downregulation. This would also improve the link to the in vivo data from Fig 6 and further boost the significance of this manuscript.

We thank the reviewer for these suggestions and now include the requested data for regulation of two miR-128 targets in neurons. Further, we show the requested 3'UTR sensor experiment with truncation mutants of Arpp21. Deletion of the RNA-binding domains completely abolishes and deletion of the C-terminal domain strongly impairs the effect of Arpp21 on the 3'UTR reporters (Supplementary Fig. 10a-d). We became interested in *Phf6* and *Msk1* as highly ranked miR-128 targets according to the TargetScan and/or Pictar predictions that each contain at least three putative miR-128 binding sites. They were also among the most downregulated mRNAs in a microarray study upon miR-128 overexpression in P19 stem cells (our unpublished observations). To satisfy the reviewer's request, we now present evidence for inhibition of endogenous *Phf6* and *Msk1* at the mRNA and protein levels in primary neuronal cultures after overexpression of miR-128 (revised Fig. 6i,j and Supplementary Fig. 11f-h). Using a lentivirus expression construct, we essentially increase the levels of miR-128 in DIV7 neurons to a level comparable to that seen in more mature neuronal cultures. DIV7 corresponds to the period of dendritic outgrowth in this *in vitro* culture system. The magnitude of the effects is comparable to what we show for the endogenous proteins in HEK293 cells.

We also confirm regulation of endogenous PHF6 and MSK1 protein levels in the same neuronal culture system using a full-length ARPP21 lentivirus

construct and an shRNA expression construct to knock down *Arpp21* (revised Fig. 6k-n). Here the effects are more modest but nevertheless consistent. This may be due to the technical difficulties involved and the greater overall complexity of gene regulation in the neuronal system. As in the cell lines mRNA levels were affected to smaller extent and changes did not reach statistical significance, as might be expected if the primary action of ARPP21 is at the level of translation (Supplementary Fig. 11j,k). Overall, we agree with the reviewer that this new data should significantly boost the paper's significance.

7) In Fig 6, the authors should include a rescue experiment by co-expression of a miR-128 sponge or miR-128 mimic, with the shArpp21 or Arpp21 respectively, thereby showing the opposing function in vivo.

We have complied with this request and now show in Supplementary Fig. 15a-c that co-expression of miR-128 blocks the ability of ARPP21 to accentuate dendritic complexity. We believe this is strong evidence that the two gene products converge on shared pathways involved in the specification of cortical neuron morphology.

8) In Fig 7, the authors could include the negative feedback loop from mature miR-128 to the Arpp21 mRNA, since miR-128 is binding and downregulating the long Arpp21 isoform (see Fig S1A). Further, I would suggest removing the terms stability and decay from the chart, since no data shown are supporting any of these conclusions.

We thank the reviewer for this suggestion and agree with the criticism. Figure 7 (Fig. 8 in the revised manuscript) has been modified accordingly.

Minor comments:

1) Check writing of the R3hdm1/2 genes or proteins in the legend to Fig 1.

Thanks for pointing this out, this was a consequence of incorporating the legend into an Illustrator file. This is corrected in the revised version.

2) Check first sentence in legend of Fig 2, there is no "host protein" for a micro-RNA. I would suggest, removing the link to the miR-128, since you are only showing data from Arpp21 and R3hdm. The authors showed in Fig 1 that those are the host genes.

We agree with the reviewer that the link to miR-128 is sufficiently introduced in Fig. 1 and we changed the term "host protein" to R3HDM1 and ARPP21.

3) There is an minor inconsistency in the data analysis in Fig 5H/I and Fig 5K/L, the analysis of the significance is different.

We thank the reviewer for the careful attention to our work. We have corrected this regrettable oversight.

Reviewer #3 (Remarks to the Author):

Here, Rehfeld and colleagues investigate the relationship between a gene that codes for the Arpp21 protein, and also codes for the intronic miR-128 microRNA. They provide evidence that Arpp21 post-transcriptionally enhances gene expression and binds to a subset of bioinformatically-predicted miR-128-targeted mRNAs. Overall, their model is that Arpp21 antagonizes miR-128 silencing capacity on select mRNAs. Moreover, they posit that this molecular mechanism regulates dendritogenesis in terminally differentiated cortical neurons the mouse brain. Overall, this is a very interesting manuscript and the data are of high quality. That being said, several points need to be addressed:

Major comments:

1) While it is quite interesting that Arpp21 enhances gene expression, and Arpp21 interacts to a certain degree with eIF4G and eIF4A, I think that the authors statement that this interaction is causative for upregulation of gene expression is a bit overreaching. One thing that I think could strengthen this argument is if the authors test whether their Arpp21 deltaC-term mutant, which cannot enhance gene expression of their reporter, is unable to co-precipitate eIF4G and eIF4A. If it can't then this would strengthen their argument. If it doesn't, then maybe eIF4G/eIF4A interaction may not play a major role in the upregulation of gene expression they observe.

This same experimental confirmation was suggested by Reviewer 2 (Point 6) and was therefore a priority for the revised version. In the revised version of Supplementary Fig. 4e we can indeed show that the C-terminal domain of ARPP21, the same domain devoid of RNA binding activity in the crosslinking assay (Fig. 4c) but with transactivation activity in the tethering assay (Fig. 3c) is necessary and sufficient for physical interaction with eIF4G and 4A. Our model for ARPP21 action via eIF4G/eIF4A therefore passes this critical test. In combination with our finding that eIF4G knockdown specifically suppresses the ability of ARPP21 to mediate transactivation, we believe our data provide a valuable foundation for future, more detailed mechanistic studies by allowing specific and testable predictions to be formulated. We have also made every effort to avoid overreach in our revised presentation of these results. However, we believe we have used multiple approaches and the resulting data set is consistent and hope sufficient for the first molecular study of this novel RNA-binding protein.

2) For IUE experiments, they need to use a separate shRNA to validate and make sure not off-target. Alternatively, the authors could co-transfect a Arpp21 rescue construct along with the shRNA to test whether this ameliorates the effect of the shRNA alone.

The result of this important control experiment is now included in the revised Supplementary Fig. 13b-d and fully supports the original conclusion. As the original shRNA negative control already argues against non-specific effects

such as RISC overload, we therefore chose the more challenging rescue experiment. An shRNA-resistant ARPP21-expression construct substantially rescued the dendritic outgrowth phenotype seen after *Arpp21* knockdown. As noted in the text, the rescue was not complete. We believe this is most likely due to experimental constraints. The first point is that it is in principle difficult to achieve physiological protein levels in this IUE experiment, which would be required to perfectly balance the knockdown. This is particularly true in this case due to the importance of temporal control, in other words providing the correct dose of ARPP21 at the right time. For technical reasons and to match the experiments we have already shown, the promoter driving the shRNA is constitutive but the NeuroD1 promoter driving the rescue construct is only active at later stages of neuronal differentiation. Despite these technical constraints, statistically significant rescue was obtained.

3) When testing various 3'UTRs as being putative miR-128 targets, the authors use a negative control reporter containing five miR-128 target sites that are perfectly complemented to miR-128 but have no poly(U)-rich sequence that can bind Arpp21. Unfortunately, this is not a proper control as this reporter will be cleaved by miR-128 via RNAi. This is due to the fact that miRNAs that are perfectly complementary to target sequences act as siRNAs. The authors need to have a control that is targeted by miR-128 via imperfect complementary. I would use a bona fide miR-128 targeted 3'UTR to model what kind of sequences they can use for their control reporter.

We use the perfectly complementary binding site reporter as a standardized positive control for all reporter assays to establish the experimental parameters for each experiment. The reviewer is correct that the mechanism of targeting differs between perfectly and imperfectly complementary miRNA binding sites so that this control should not be used for mechanistic comparisons. We had omitted this important control due to space constraints and have now performed an additional test to meet this criticism.

We used two naturally occurring 3'UTRs found in the mRNAs for two related protein kinases, MSK2 and MSK1. Whereas the 3'UTRs for both are subject to targeting by miR-128, only the 3'UTR of *Msk1* was bound by ARPP21 in the iCLIP data (~23 crosslinking events per TPM; TPM=5.2; Fig. 6b; Supplementary Table 6c). *Msk2* was not recovered by iCLIP despite similar expression levels in the iArpp21 cells (Supplementary Table 6b,c; TPM=5.9). Neither the baseline expression nor the magnitude of miR-128 mediated suppression of an *Msk2*-3'UTR reporter construct was affected by ectopic ARPP21 (revised Supplementary Fig. 8c).

We also noticed a species-specific variant in the 3'UTR of the *Upf1* mRNA that allowed a further test of our model. *Upf1* is a predicted and experimentally validated target for miR-128 in mouse and human³. The mouse 3'UTR contains a unique extended uridine-rich region (72 uridines out of 88nt in total; Supplementary Fig. 9a) not present in the human. Deletion of this polyU stretch in a mouse *Upf1* 3'UTR construct strongly diminished the ability of ARPP21 to stimulate expression of the reporter without significantly affecting basal reporter expression or miR-128 targeting (Supplementary Fig. 9b,c).

Together, we think the inclusion of this new data should ameliorate the reviewer's concerns and strengthen the manuscript.

4) A major limitation of the CLIP-Seq data obtained is that it is all carried out in a non-neuronal cell line (i.e. HEK-293 cells), yet the main impact points of the study investigate Arpp21 function in the nervous system. The author should validate a select population of their Arpp21-targeted mRNAs in either primary neuronal cultures, or neuronal cell lines such as N2a or SH-Y5Y. This would help to confirm that their iCLIP datasets do not just occur in the context of HEK-293 cells.

We agree with Reviewer 3, Reviewer 2 (Point 6) and the editor that an increased focus on neuronal gene expression would strengthen our manuscript. We therefore elected to perform the more challenging but also more physiologically relevant experiments in primary neurons rather than neuron-like cell lines. As detailed in the response to Reviewer 2 Point 6 we used lentiviral-mediated ARPP21 overexpression to document increased expression at the protein level for two significant ARPP21 targets: PHF6 and MSK1 (Fig. 6 k,m). The effects are more modest but nevertheless highly consistent with the iCLIP and the analogous expression assays conducted in HEK293 cells. The lower general magnitude of the effects may not be surprising due to the technical difficulties involved and the greater overall complexity of gene regulation in the neuronal system.

5) The authors posit that Arpp21 expression in neurons is critical for regulating dendrite arborisation in part by opposing miR-128 action. Unfortunately, the data support a genetic linkage, but not a direct mechanistic link between miR-128 and Arpp21. Can the authors provide any evidence that Arpp21 is acting on miR-128-targeted mRNAs in neurons?

We have already detailed the new experimental data that support the relevance of our original findings for neuronal gene regulation (Reviewer 2 Point 6, Reviewer 3 Point 4). We can indeed show that miR-128 and ARPP21 have inverse effects on the regulation of PHF6 and MSK1 in cortical neurons. We believe that MSK1 is a significant player in the known ability of miR-128 to modulate the MAPK signaling pathway in neurons. We have previously shown that regulation of PHF6 by miR-128 is involved in the regulation of cortical neuron migration and dendritic arborization. In our view, this evidence for co-regulation of key miR-128 target mRNAs is evidence for a mechanistic rather than merely a genetic link. This reasoning is born out by the phenotypes we observe upon manipulation of either miR-128 or ARPP21 *in vivo*: either miR-128 overexpression⁹ or ARPP21 knockdown reduces dendritic complexity; ARPP21 overexpression increases dendritic complexity. This is consistent with our previous report that co-electroporation of PHF6 rescues the dendritic defects seen upon miR-128 overexpression⁹.

The current manuscript describes the interplay between miR-128 and its host gene and identifies key regulatory pathways and nodes downstream of the two genes. We feel that a detailed mechanistic investigation of how PHF6, an epigenetic regulator about which little is known, controls dendritic growth is

beyond the scope of this paper. We nevertheless believe that the extensive overlap we describe in the regulatory targets for miR-128 and ARPP21, and their enrichment for regulatory pathways relevant for neuronal functions, is sufficient evidence for a novel kind of regulatory feedback loop. We know of no reason to think that the mechanisms used by either miR-128 or ARPP21 to control gene expression are fundamentally different in cell lines or neurons. We now support this assumption by replicating key cell line results in neurons wherever possible and hope we have adequately addressed the reviewer's concerns.

References

1. Huppertz I, *et al.* iCLIP: protein-RNA interactions at nucleotide resolution. *Methods* **65**, 274-287 (2014).
2. Tan CL, *et al.* MicroRNA-128 governs neuronal excitability and motor behavior in mice. *Science* **342**, 1254-1258 (2013).
3. Bruno IG, *et al.* Identification of a microRNA that activates gene expression by repressing nonsense-mediated RNA decay. *Mol Cell* **42**, 500-510 (2011).
4. Lou CH, *et al.* Posttranscriptional control of the stem cell and neurogenic programs by the nonsense-mediated RNA decay pathway. *Cell Rep* **6**, 748-764 (2014).
5. Gregersen LH, *et al.* MOV10 Is a 5' to 3' RNA helicase contributing to UPF1 mRNA target degradation by translocation along 3' UTRs. *Mol Cell* **54**, 573-585 (2014).
6. Meijer HA, *et al.* Translational repression and eIF4A2 activity are critical for microRNA-mediated gene regulation. *Science* **340**, 82-85 (2013).
7. Schwanhausser B, *et al.* Global quantification of mammalian gene expression control. *Nature* **473**, 337-342 (2011).
8. Chapat C, *et al.* Cap-binding protein 4EHP effects translation silencing by microRNAs. *Proc Natl Acad Sci U S A* **114**, 5425-5430 (2017).
9. Franzoni E, *et al.* miR-128 regulates neuronal migration, outgrowth and intrinsic excitability via the intellectual disability gene Phf6. *Elife* **4**, (2015).

Reviewers' comments:

Reviewer #2 (Remarks to the Author):

The authors have successfully strengthened the connection to neuronal functions in the revised manuscript. Several key experiments have now been performed in neurons and essential controls have been added. The only potential pitfall of this significantly improved revision is that the authors did not consider including data on the corresponding RNA levels for the luciferase and tethering assays.

Reviewer #3 (Remarks to the Author):

Reviewer comment: A major limitation of the CLIP-Seq data obtained is that it is all carried out in a non-neuronal cell line (i.e. HEK-293 cells), yet the main impact points of the study investigate Arpp21 function in the nervous system. The author should validate a select population of their Arpp21-targeted mRNAs in either primary neuronal cultures, or neuronal cell lines such as N2a or SH-Y5Y. This would help to confirm that their iCLIP datasets do not just occur in the context of HEK-293 cells.

Author response: We agree with Reviewer 3, Reviewer 2 (Point 6) and the editor that an increased focus on neuronal gene expression would strengthen our manuscript. We therefore elected to perform the more challenging but also more physiologically relevant experiments in primary neurons rather than neuron-like cell lines. As detailed in the response to Reviewer 2 Point 6 we used lentiviral-mediated ARPP21 overexpression to document increased expression at the protein level for two significant ARPP21 targets: PHF6 and MSK1 (Fig. 6 k,m). The effects are more modest but nevertheless highly consistent with the iCLIP and the analogous expression assays conducted in HEK293 cells. The lower general magnitude of the effects may not be surprising due to the technical difficulties involved and the greater overall complexity of gene regulation in the neuronal system.

Reviewer response: The only missing pieces of data that the authors should test is that PHF6 and MSK1 mRNAs are actually targeted by miR128 in neurons. Can the authors show that the 3'UTRs for these genes are the same in primary neurons as in the HEK-293 cells? At the very least, do they still maintain miR-128 and ARPP21 binding sites? Can ARPP21 immunoprecipitate these mRNAs from primary neurons (RT-qPCR)?

Response to Reviewer 3

“The only missing pieces of data that the authors should test is that PHF6 and MSK1 mRNAs are actually targeted by miR128 in neurons. Can the authors show that the 3’UTRs for these genes are the same in primary neurons as in the HEK-293 cells? At the very least, do they still maintain miR-128 and ARPP21 binding sites?”

We appreciate the Reviewer’s comment and the attention to detail involved. In response, we would first like to remind the Reviewer that we do show that the endogenous *Msk1* and *Phf6* transcripts are downregulated by exogenous miR-128 in primary cortical neurons (Fig. 6i-j). We also show that at the protein level MSK1 and PHF6 expression is sensitive to ARPP21 in primary cortical neurons (Fig. 6k-n). As an additional point, although the iCLIP results demonstrate ARPP21 binding to the human HEK293 transcripts, all of the reporter assays used to verify these results were performed using the major annotated 3’UTRs from mouse. With the interesting exception of *Upf1* (murine specific poly-U stretch, see Supplementary Fig. 9), we did not observe relevant differences in 3’UTR binding sites for either miR-128 or ARPP21 among the transcripts we used for our verification experiments.

To better illustrate this point and to ameliorate the Reviewer’s concerns we now compile RNA-Seq data from publicly available expression datasets that allow direct comparison of transcript splicing and 3’UTR utilization from HEK-293 cells and mouse cortical projection neurons at different developmental time points (Rybak-Wolf *et al.*, 2014; Molyneaux *et al.*, 2015). This analysis is presented in a new Supplemental Figure (Supplementary Fig. 12). It directly shows that the 3’UTRs of both the *Msk1* and the *Phf6* transcripts are analogous in cortical neurons and HEK-293 cells. At the sequence level, this similarity includes the conserved poly-uridine rich ARPP21 binding regions and all of the predicted (and experimentally validated) miR-128 binding sites. It is worth noting an additional point of interest. The RNA-Seq data show significant downregulation of both transcripts from embryonic day 16 to postnatal day 1. This is perfectly consistent with our model: we show in Supplementary Fig. 1c that miR-128 expression increases about 5-fold between these developmental timepoints. This is also in agreement with our published characterization of miR-128 expression in the upper layer cortical projection neurons by *in situ* hybridization (Franzoni *et al.*, 2015).

With this additional evidence, we believe we have successfully and very specifically satisfied the Reviewer’s request. We hope the Reviewer is convinced that we have carefully and systematically used a detailed characterization of ARPP21 binding obtained with a state-of-the-art screen in HEK-293 cells to identify and functionally verify candidate mRNAs of biological relevance for miR-128 and ARPP21 in neurons and cortical development.

“Can ARPP21 immunoprecipitate these mRNAs from primary neurons (RT-qPCR?)?”

We would like to start by explicitly defending our decision to perform iCLIP in a cell line using carefully controlled and stringent conditions. The advantages of this approach for the description of direct, high resolution RNA-protein interactions compared to RIP (RNA co-immunoprecipitation) are well established (Ule *et al.*, 2005; Huppertz *et al.*, 2014). However, we agree with the Reviewer that RIP

experiments can be useful for the validation of iCLIP results. To comply with this request, we performed RIP analysis of endogenous ARPP21 from embryonic mouse cortical tissue. The results are presented in a new Supplementary Figure (Supplementary Fig. 13) and fully support our prediction that the *Msk1* and *Phf6* transcripts are bound by ARPP21 *in vivo*. Compared to the RIP experiments we have already performed in HEK-293 cells (Supplementary Fig. 7a-c), the relative enrichment of the two transcripts of interest compared to control is less dramatic. This difference is not unexpected and is most likely due, at least in part, to the limitations of our self-made ARPP21 antibody (no superior alternative antibodies are available to the best of our knowledge) and of course to possible limitations of the RIP approach when applied to complex biological samples. We hope that we have now provided “the only missing pieces of data” to the Reviewer’s satisfaction. We thank the Reviewer for the constructive criticism and think that this effort has strengthened our manuscript.

References

- Franzoni, E., Booker, S.A., Parthasarathy, S., Rehfeld, F., Grosser, S., Srivatsa, S., Fuchs, H.R., Tarabykin, V., Vida, I. & Wulczyn, F.G. (2015) miR-128 regulates neuronal migration, outgrowth and intrinsic excitability via the intellectual disability gene *Phf6*. *eLife*, **4**.
- Huppertz, I., Attig, J., D’Ambrogio, A., Easton, L.E., Sibley, C.R., Sugimoto, Y., Tajnik, M., König, J. & Ule, J. (2014) iCLIP: Protein–RNA interactions at nucleotide resolution. *Methods*, **65**, 274-287.
- Molyneaux, B.J., Goff, L.A., Brettler, A.C., Chen, H.H., Hrvatin, S., Rinn, J.L. & Arlotta, P. (2015) DeCoN: genome-wide analysis of *in vivo* transcriptional dynamics during pyramidal neuron fate selection in neocortex. *Neuron*, **85**, 275-288.
- Rybak-Wolf, A., Jens, M., Murakawa, Y., Herzog, M., Landthaler, M. & Rajewsky, N. (2014) A variety of dicer substrates in human and *C. elegans*. *Cell*, **159**, 1153-1167.
- Ule, J., Jensen, K., Mele, A. & Darnell, R.B. (2005) CLIP: a method for identifying protein-RNA interaction sites in living cells. *Methods*, **37**, 376-386.

REVIEWERS' COMMENTS:

Reviewer #3 (Remarks to the Author):

I appreciate the authors' response to my lingering concerns, which they have successfully alleviated.

I support the publication of their manuscript in Nature Communications.

• Reviewer #3 (Remarks to the Author):

“I appreciate the authors' response to my lingering concerns, which they have successfully alleviated.

I support the publication of their manuscript in Nature Communications.”

We thank each of the Reviewers and in this case particularly Reviewer 3 for their time and the substantial effort they offered on our behalf and on the behalf of our readers.